# Is This Tracker On?
# A Benchmark Protocol for Dynamic Tracking

**Ilona Demler**\* **Saumya Chauhan**\* **Georgia Gkioxari**
California Institute of Technology
{idemler, schauhan, georgia}@caltech.edu

## Abstract

We introduce ITTO, a challenging new benchmark suite for evaluating and diagnosing the capabilities and limitations of point tracking methods. Our videos are sourced from existing datasets and egocentric real-world recordings, with high-quality human annotations collected through a multi-stage pipeline. ITTO captures the motion complexity, occlusion patterns, and object diversity characteristic of real-world scenes – factors that are largely absent in current benchmarks. We conduct a rigorous analysis of state-of-the-art tracking methods on ITTO, breaking down performance along key axes of motion complexity. Our findings reveal that existing trackers struggle with these challenges, particularly in re-identifying points after occlusion, highlighting critical failure modes. These results point to the need for new modeling approaches tailored to real-world dynamics. We envision ITTO as a foundation testbed for advancing point tracking and guiding the development of more robust tracking algorithms.

 **Data & Code:** github.com/ilonadem/itto
 **Data & Dataset Card:** huggingface.co/datasets/demalenk/itto-dataset
 **Project Website:** https://glab-caltech.github.io/ITTO/

## 1 Introduction

Motion underlies the physical world. Understanding how objects move over time is critical for downstream applications in robotics, video-scene understanding, and human-computer interaction. In this paper we investigate this through point tracking in videos, or estimating point locations across frames containing dynamic objects. This is a hard problem; real-world settings contain complex non-rigid body motions, objects disappear and reappear again in varying locations, and cameras move during data capture. A plethora of methods have been proposed to tackle these challenges, from feed-forward transformers that process temporal sequences [17, 8, 3, 9, 8, 21] to dynamic memory banks that encode motion and appearance history [1]. Yet, performance is reported on simplistic benchmarks, hindering progress and preventing a clear understanding of the methods' failure modes.

Existing real-world datasets such as the popular TAP-Vid Davis [7] contain relatively simple scenes: typically one or two moving foreground objects, simple camera motions, few occlusions, and of short duration ($\leq$ 100 frames per video). TAP-Vid Kinetics, another popular benchmark sourced from YouTube videos, has spatio-temporally sparse annotations with 88% of tracks being slow or static. DynamicReplica [18] and PointOdyssey [37] are longer range and of increased motion complexity, but contain only synthetic videos that fall far from the nature and diversity of motion in the real world. Crucially, metrics reported on all these datasets fail to capture why and when models fail; an inability to track through occlusions is entangled with failure to bind to specific object features and inability to distinguish between similar points. However, understanding when and why a tracking model breaks is essential in improving its performance and deploying it in the real world.

---

\*Equal contribution.

39th Conference on Neural Information Processing Systems (NeurIPS 2025) Track on Datasets and Benchmarks.

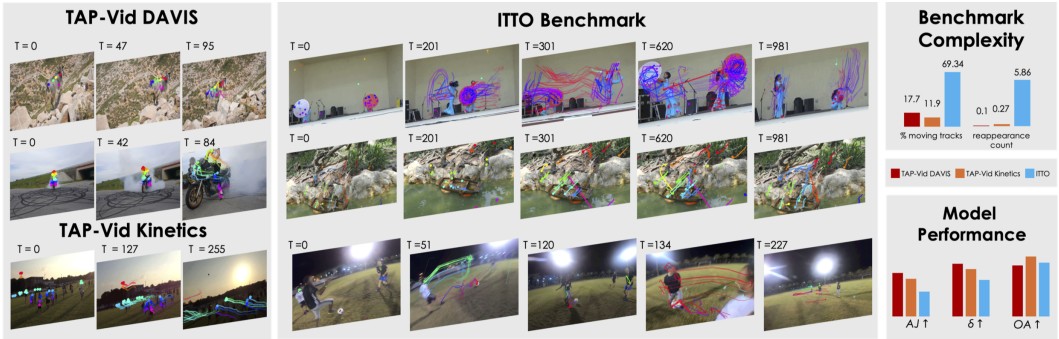

Figure 1: **Overview of ITTO benchmark vs. other benchmarks**. ITTO has more objects per video, highly deformable objects (such as umbrella opening and closing in top row) and all phenomena of the real world (significant % of moving tracks, and high track reappearance count). SOTA tracking methods perform worse on ITTO compared to the existing benchmark; e.g. CoTracker3 [17] performance drops from 61.8 to 35.5 for Average Jaccard (AJ) and from 76.1 to 47.1 for $\delta$.

In our work we consider the problem of tracking over explicit complexity axes, which we define as videos with substantial changes in appearance, motion patterns, high object disappearance/reappearance, and significant camera displacement. To this end, we present a new benchmark, ITTO, which consists of real-world videos sourced from a variety of monocular and egocentric datasets. We show that ITTO covers the diversity and nuance of the real-world and is significantly more challenging than all existing benchmarks, with $14\times$ as many per-track reappearances, $4.5\times$ as many moving points, and $2\times$ as many tracked objects per video. We show an overview of ITTO together with a comparison to state-of-the-art tracking benchmarks in Figure 1.

We intend for ITTO to be used as a testing suite for diagnosing the strengths and limitations of any tracking model in real-world scenarios. We use ITTO to conduct a rigorous analysis of current state-of-the art 2D and 3D trackers, and report performance along each axis of motion complexity. First, we find that standard track performance metrics degrade significantly: CoTracker3 [17], the best performing current tracking model, achieves 64.4% Average Jaccard on TAP-Vid Davis and only 42.0% on ITTO. While all methods are able to track stationary and background points near-perfectly, they are ill-equipped to track through higher rates of frame-to-frame track motion, suggesting an implicit stationary bias. Additionally, methods struggle to track beyond their native context windows and exhibit catastrophic tracking forgetting, unable to recover tracks once they reappear from an occlusion. Tracking through re-appearance is not emphasized in many benchmarks and thus methods are not designed to handle such phenomena. This suggests that novel formulations of memory and dynamic tracking are needed to augment existing models with the ability to re-identify tracks.

In short, in this work we provide the following contributions:

1. ITTO, a long-range and real-world tracking benchmark with novel motion complexity that is missing in current tracking evaluation benchmarks.

2. A rigorous evaluation protocol and assessment of current state-of-the-art 2D and 3D tracking models along defined axes of complexity.

3. A plug-and-play data annotation pipeline for collecting high-quality video track annotations.

## 2 Related Work

**Tracking methods.** Video motion modeling has evolved across multiple paradigms. Optical flow approaches estimate pixel displacements between frames but struggle with occlusions and long-term tracking [22, 14, 16, 28, 36, 29]. To address these limitations, Sand and Teller [27] proposed the tracking-any-point task, modeling motion via long-range point trajectories. Recent methods use temporal transformers, training on pseudolabels, and feature matching [12, 1, 17, 8, 13, 3, 32, 34, 21]. Extensions to 3D point tracking leverage depth estimation and multi-view geometry [23, 31, 34], test-time optimization [33], or retraining on joint 3D features [32]. Our work evaluates both 2D and 3D tracking approaches on 2D annotations.

**Point tracking benchmarks.** A key challenge in tracking-any-point tasks is the scarcity of annotated real-world datasets. Evaluations primarily use TAP-Vid Davis and TAP-Vid Kinetics [7]. TAP-Vid Davis features 30 videos from the DAVIS dataset [25], with 34-104 frames and just 26.3 points per video on average, mostly on single foreground objects followed closely by the camera, resulting in minimal parallax and viewpoint variation. While TAP-Vid Kinetics offers more scale (1,144 YouTube videos from Kinetics [19]), many are composite videos that also focus on foreground objects. Most tracking models train on Kubric [11], a large-scale synthetic dataset of falling indoor objects, and PointOdyssey [37], which introduces camera and asset motions collected from real-world datasets, but these are far from natural and primarily used for training. Several synthetic benchmarks cover longer sequences [18, 7, 37], but they similarly fail to capture real-world visual and motion complexity. We discuss the limitations of these benchmark datasets further in Section 3.1.

**Object tracking datasets.** Video-object segmentation (VOS) focuses on generating consistent object masks across all video frames [35], and datasets for training and evaluation include videos with complex objects and motion patterns [2, 30, 15, 24, 26, 35, 25, 20]. MOSE presents a VOS dataset focused on cluttered scenes in videos up to 20 seconds long [6] and LVOS [35] focuses on long temporal ranges of up to 334 minutes. Another source of video datasets, not targeting point or object tracking, are egocentric videos [4]. Ego4D [10] is a large-scale 2,670 hour egocentric video dataset spanning daily-life head-mounted activities from sports to cooking. We leverage these videos due to their complexity and importance of visual domain in our benchmark.

# 3 The ITTO Tracking Benchmark

In this section, we introduce ITTO, a tracking benchmark designed to capture the motion complexity of the real-world. Grounded in real-world scenes, ITTO features multiple objects undergoing diverse motions, including repeated appearance and disappearance, rapid position changes, and non-rigid motions. Individual points on these objects also experience occlusions and reappearances due to object interactions and self-occlusions. We quantitatively compare ITTO to existing benchmarks and show that it is a much more challenging setting for real-world tracking.

## 3.1 Comparison to existing tracking benchmarks

Our ITTO benchmark captures key aspects of motion that are underrepresented in existing real-world and synthetic tracking datasets. Specifically, it has greater complexity along the following dimensions: *static points*, *reappearance count*, *track duration*, and *occlusion rate*, which are critical for modeling real-world motion. We quantitatively analyze these characteristics and summarize results in Table 1. We define static points as points with frame-to-frame displacement $\leq 1.5\%$ the frame diagonal, a perceptually reasonable cutoff below which points appear visually static (e.g. for a $512 \times 512$ video this is equivlaent to a 10.9 pixel displacement). In the Appendix we provide track motion information showing ITTO's increased motion from prior benchmarks. Reappearance count refers to the number of times a point transitions from occluded to visible within a video. Occlusion rate is the proportion of frames during which a point is occluded. We also report the average number of objects per video as a measure of scene complexity, using provided ground-truth masks for TAP-Vid Davis and estimating object counts for TAP-Vid Kinetics by querying Molmo [5] on frames where tracks first appear.

From Table 1, we observe a stationary bias in real-world benchmarks: $82\%$ of tracks in Davis and $88\%$ in Kinetics are static under our definition. This reduces tracking to handling minor motion deformations, with most of the "work" effectively offloaded to the video capture process, even when the objects themselves move in complex ways. By contrast, only 30.1% of tracks in ITTO are static, significantly increasing complexity. Although Davis and Kinetics exhibit moderate occlusion (31% & 41%, respectively), their low reappearance counts (0.10 & 0.2) indicate that disappearing tracks rarely return, and thus they do not test a model's ability to track through occlusions. Conversely, ITTO tracks exhibit heavy occlusions (58.1%) with a high track reappearance count (5.86); our dataset requires that models re-identify a track once it disappears. Most Davis and Kinetics videos also contain only one moving foreground object, with 3.3 and 5.1 annotated objects per video. ITTO videos more accurately reflect the clutter of the real world, with 10.9 annotated objects per video. This increased scene complexity is paired with longer tracks, averaging 168 frames (ranging from 34-1014), longer than all real-world datasets and well beyond most model input windows.

Table 1: **ITTO vs. Other Benchmarks.** We compare ITTO to existing tracking benchmarks and report motion complexity characteristics. ITTO contains significantly fewer static points, a higher track reappearance count, and higher number of objects per video than all current benchmarks. Its average track duration is longer than all real-world datasets and on par with synthetic ones.

| Dataset | Real? | Num Tracks | Static Points | Reapp Count | Objs per Video | Track Duration | Num Videos | Occlusion Rate | Total Frames |
|---|---|---|---|---|---|---|---|---|---|
| TAP-Vid Davis | ✓ | 650 | 82.3% | 0.10 | 3.3 | 44.0 | 30 | 31.0% | 1,999 |
| TAP-Vid Kinetics | ✓ | 28,600 | 88.1% | 0.27 | 5.1 | 146.0 | 1144 | 40.6% | 286,000 |
| TAP-Vid RGB Stacking | ✗ | 12,500 | 94.2% | 0.08 | – | 167.0 | 50 | 35.0% | 12,500 |
| Dynamic Replica | ✗ | 6,000 | 99.0% | 1.02 | – | 215.4 | 20 | 28.0% | 6,000 |
| ITTO | ✓ | 1,373 | 30.7% | 5.86 | 10.9 | 221.6 | 72 | 58.1% | 11,449 |

Table 2: **ITTO vs TAP-Vid Davis object complexity** We report the object mask complexity of ITTO compared to TAP-Vid Davis. Video and object BOR denotes bounding box occlusion rate averaged per-frame and per-video. ITTO videos contain more objects and a higher BOR, on both the object and video level.

| Dataset | Video BOR Mean | Video BOR Std | Obj BOR Mean | Obj BOR Std | Num Obj Masks |
|---|---|---|---|---|---|
| TAP-Vid Davis | 0.09 | 0.06 | 0.06 | 0.06 | 2.56 |
| ITTO | 0.22 | 0.13 | 0.48 | 0.26 | 10.90 |

ITTO is also more challenging compared to synthetic benchmarks, which typically have over 90% static points, reappearance counts below 1.02, and similar track durations. Crucially, the visual domain of these datasets is completely different and the motions in them unnatural, making them less informative for real-world generalization. We note that Kubric [7] and PointOdyssey [37] are other popular synthetic datasets, but they exhibit similar limitations and we omit them from our analysis as they are not benchmarks but rather used for training.

In section 4 we explore how the complexity of ITTO is reflected by degrading performance across all modern trackers. We also conduct a more comprehensive analysis of failure modes along each of our defined axes of motion, and provide a discussion of ideas for future model development.

## 3.2 Benchmark collection & annotation protocol

In this section, we describe the sourcing, collection, and annotation process for our ITTO benchmark. We want our videos to include complex, real-world motion across diverse settings, using both monocular and egocentric video. We source the monocular portion from Video-Object Segmentation datasets MOSE [6] and LVOS [35], which provide ground-truth mask annotations. Selected videos must satisfy at least two of the following conditions: BOR > 0.2, more than two object disappearances, and a minimum length of 80 frames. BOR, formally defined in the Appendix, is the bounding box occlusion rate, or the fraction of each object's bounding box that is occluded in a frame. Object BOR is $0.22 \pm 0.13$ at the video level and $0.48 \pm 0.26$ at the object level in ITTO, compared to $0.09 \pm 0.06$ and $0.06 \pm 0.06$ for TAP-Vid Davis (Table 2). Thus, objects in ITTO are significantly more occluded than in TAP-Vid.

For the egocentric portion of our dataset, we source videos from Ego4D [10], a large-scale 2,670 hour egocentric video dataset of daily-life head-mounted activities. We generate pseudo-mask labels by querying Molmo [5] with semantic concepts present in each video, which produces coordinate positive and negative prompts that we pass into SAM2 [26] to produce object masks. This ensures a sufficient number of tracked objects per video (10.9 in ITTO vs. 3.3 in TAP-Vid Davis). We exclude the egocentric portion from the reported BOR metrics due to the lack of ground-truth segmentations. However, these videos remain challenging due to complex camera motion and frequent occlusions, reflected by low model performance reported in Section 4.

**Annotation pipeline.** As the videos in ITTO are challenging and their motions complex, we found that human annotations cannot be collected in a single step. We employ a two-phase approach:

1. **Coarse tracks**: We generate coarse track annotations through Amazon Mechanical Turk (MTurk), providing annotators with a reference frame containing the point query and a query frame side by side. They are asked to label the correct pixel in all query frames throughout the video or indicate when the track is occluded.

2. **Video track refinement**: We upload these coarse MTurk annotations to our own video tool that we send out to a team of carefully selected UpWork annotators. The tool allows users play the video and accompanying track query forward and backward in time. Users are prompted to "refine" the track until it is spatio-temporally consistent.

We provide the setup of each annotation stage in the Appendix. For the most challenging videos (based on human assessment), we add a third refinement step, re-querying the final phase 2 annotations on the video annotation tool. We note that since we employ the same team of annotators for phase 2, the annotators gain expertise and skill in producing high-quality track annotations. Intuitively, we can view this first stage as a form of track "identification" – by prompting annotators one image at a time, we ask them to re-identify the pixel in later frames – and the second stage is a form of track refinement that ensures temporal consistency throughout the video. Another benefit of our data collection setup is that there is no algorithmic bias in our annotation generation as we do not deploy algorithmic track initialization and propagation as done in TAP-Vid [7].

**Ease of tool use.** In all phases of annotation collection, we provide bounding-box information for point queries that correspond to object masks. This is particularly critical in phase 1, as some of our videos contain repeated objects (e.g., many sheep running in the woods). In order to make the second stage maximally efficient, we provide keyboard controls that let users seamlessly correct the tracks as they play through the video. We include further details about the tool setup in the Appendix.

**Query point selection.** Current tracking benchmarks such as TAP-Vid rely on human annotators to pick which pixels to track, which results in an inherent bias towards salient points. Our query point selection process is algorithmic. We first identify moving objects in a video, determine the number of points to sample per object, and then sample gradient-based, random, and background points. For the monocular portion of our dataset (sourced from MOSE and LVOS), we utilize available ground-truth object mask information. For the egocentric portion (Ego4D), we query Molmo [5] with semantic concepts to produce positive and negative query prompts for SAM2 [26], generating pseudo-mask labels. Each object's point count scales with its area relative to the frame, while background points are capped at <20% of the total. For the monocular portion of our dataset which we source from MOSE and LVOS, we take advantage of available ground-truth object mask information. For the egocentric portion of our dataset, we query Molmo [5] with semantic concepts such as "head", "hand", "ball", or "shoe", which produces positive and negative query prompts that we pass into SAM2 [26] to produce pseudo-mask labels. For all videos in our dataset, each object's point count scales with its area relative to the frame, while background points are capped at $< 20\%$ of the total.

Concretely, for each video we first distinguish moving objects (whose mask-centroid travels more than $0.1\%$ the frame diagonal, a stricter definition of motion than in Table 3.1) from stationary ones, which we treat as background. We allocate $p = \max\left(1, \left\lfloor 50 \times \frac{\text{area}_{\text{obj}}}{\text{area}_{\text{frame}}} \right\rfloor\right)$ query points per query frame for each moving object. As the videos in LVOS have a higher resolution and the SAM2-generated masks in Ego4D are noisy, we additionally cap $p$ at 5 and 10 points per-frame per-object. To produce gradient-based query points, we select the top $p$ pixels by Sobel gradient magnitude ($\tau > 20$) within each mask interior (10 px from mask boundary). Random-based points are sampled uniformly from the remaining interior pixels. Background queries are drawn uniformly from non-moving regions. For all videos, queries are sampled at frame 0. Queries are sampled at frame 0 for all videos, every 100 frames for LVOS, and a $\frac{1}{4}$, $\frac{1}{2}$, and $\frac{3}{4}$ of the video length for Ego4D. This results in a variation in the number of point tracks per video proportional to occlusion complexity. We show qualitative examples from our dataset in Fig. 1.

**Validation of annotation quality.** To validate the accuracy of our generated annotations, we run our annotation pipeline on a subset of Kubric, synthetic videos with ground-truth point tracks. We found that 18.25% of points were within 2 pixels of ground truth with a mean annotation error of 9.75 pixels after the MTurk step (phase 1), and 88.3% of points were within 2 pixels after the video annotation step (phase 2) with a mean track error of 1.32 pixels. For comparison, the TAP-Vid annotation pipeline produces 80% of tracks within 2 pixels on Kubric data. Our pipeline thus produces a "coarse" set of track annotations in the MTurk step, which are then refined to near ground-truth accuracy.

**Ethics of annotation collection** ITTO contains videos of human and animal subjects sourced from publicly-available datasets cleared for research use. MTurk workers voluntarily selected our task

from an available pool and were compensated per-track annotation. Upwork annotators, who served as expert annotators during the data collection process, were paid by hourly contracts with clear instructions and ongoing feedback. We screened and onboarded them with Kubric annotation tasks.

# 4    Evaluation and Metrics

Our ITTO benchmark extends prior tracking datasets by explicitly spanning multiple axes of motion complexity (summarized in Table 1). We assess state-of-the-art trackers on ITTO and dissect their results with a new metric suite that breaks down performance into motion-complexity tiers. This evaluation reveals critical failure modes in current methods, especially under occlusion, reappearance, and high-motion conditions, as well as a catastrophic inability to recover from tracking errors. We thus position ITTO not just as a benchmark, but also as a diagnostic tool that illustrates the pressure points of existing models and identifies future directions for achieving robust, real-world tracking.

**Point tracking methods.**    We explore how ITTO's novel characteristics affect current model performance by evaluating ten state-of-the-art architectures that report the best performance on the TAP-Vid Davis and Kinetics benchmarks. These models span various strategies: transformer-based architectures deforming stationary track initializations [17], coarse-to-fine approaches using feature cross-correlations [9, 3], models leveraging large-scale data via student-teacher training on pseudo-labels [8, 17], and deformable transformers formulating tracking as object detection [21].

We also consider 3D trackers, many of which augment 2D tracking with monocular depth estimators [23, 34] or are built to operate on RGB-D videos [31], which we produce via pretrained depth prediction models. We omit VGGT [32] as they have not released model weights. As ITTO videos contain significant motion and repeated objects, we find that track initialization strategies and feature matching play a critical role in model performance, which we explore below.

## 4.1    Partitioned metrics: axes of motion complexity

To better understand how individual models perform on ITTO and investigate how robust they are to different aspects of motion complexity, we propose a new delineation of track metrics that we hope the community adopts when reporting tracking performance. We partition ITTO tracks along two axes: track motion and reappearance frequency. We find that these definitions provide us with a quantitative measure of how robust a model is to these aspects of motion difficulty, and give us a measure of potential breaking points for real-world deployment.

*Track motion.* Tracks are split based on average frame-to-frame-motion, categorized by thresholds of $[0.5\%, 1.5\%, 5\%]$ relative to the frame diagonal. Tracks that have $< 0.5\%$ motion are effectively static. This gives a quantitative measure of how well trackers can handle rapid movements.

*Reappearance Frequency.* This categorization investigates how well models are able to keep tracking through occlusions. Tracks are divided into three equal-sized bins based on the number of times the track reappears (transitions from occluded to unoccluded) throughout the video. Notably TAP-Vid DAVIS and Kinetics do not contain this information, as their tracks contain one or fewer reappearances on average, meaning they have very few tracks in the higher reappearance categories.

We report standard tracking metrics following [7], averaged over all tracks and per motion partition: average points within Tadelta ($\delta$), Average Jaccard (AJ), and occlusion accuracy (OA). $\delta$ (also referred to as $\delta_{avg}^x$ in other works), measures the average fraction of points that are within $[1, 2, 4, 8, 16]$ pixels from ground truth visible points. AJ measures the fraction of true positives divided by the true positives, false positives, and false negatives combined, where true positives are points within a specified pixel threshold of ground truth visible points, false positives are points predicted visible but are either outside of threshold or ground truth occluded, and false negatives are ground truth visible points that are predicted visible or outside of threshold. AJ averages this fraction over the same $[1, 2, 4, 8, 16]$ pixel thresholds as $\delta$. OA measures the classification accuracy of the occlusion predictions. Higher values indicate better model performance for all metrics. In addition to track motion and reappearance frequency, we also propose the Pairwise Distance Variance (PDV) as a complementary metric that captures the spatial coherence of tracks belonging to the same object mask throughout the duration of the video. We include a discussion of the PDV in Appendix 6.2.

Table 3: **Performance of SOTA tracking models on ITTO**. We report standard tracking metrics, Average Jaccard (AJ), average points within delta ($\delta$), and occlusion accuracy (OA) for SOTA models on our ITTO benchmark; higher values indicate better performance. The horizontal line delineates 2D and 3D methods. For all models, average performance on all metrics drops significantly compared to TAP-Vid. We also report performance along motion complexity tiers of track motion and number of visible consecutive frame segments per track, and see further significant degradation for high track motion and visible track segment counts.

| | Average | | | Track Motion | | | | | | | | | | | | Reappearance Frequency | | | | | | | | |
| | | | | [0%, 0.5%) | | | [0.5%, 1.5%) | | | [1.5%, 5%) | | | [5%, 100%) | | | [0, 1) | | | [1, 3) | | | [3, 1000) | | |
| | AJ | $\delta$ | OA | AJ | $\delta$ | OA | AJ | $\delta$ | OA | AJ | $\delta$ | OA | AJ | $\delta$ | OA | AJ | $\delta$ | OA | AJ | $\delta$ | OA | AJ | $\delta$ | OA |
|---|---|---|---|---|---|---|---|---|---|---|---|---|---|---|---|---|---|---|---|---|---|---|---|---|
| CoTracker3 online [17] | 35.3 | 47.0 | **77.6** | **55.9** | **68.4** | 86.1 | **35.3** | **46.7** | 76.9 | **25.3** | **33.5** | **74.4** | **13.7** | **22.1** | 75.7 | **50.1** | **60.5** | 88.7 | 23.7 | 33.5 | 72.4 | 13.7 | 21.5 | 65.9 |
| CoTracker3 offline [17] | 33.7 | 45.0 | 76.9 | 55.0 | 65.4 | **87.7** | 33.3 | 42.0 | 76.1 | 21.0 | 27.7 | 72.1 | 11.2 | 16.6 | **76.2** | 49.3 | 58.3 | **89.1** | 21.2 | 29.1 | 71.3 | 10.9 | 16.8 | 63.3 |
| LocoTrack [3] | **36.0** | **52.8** | 75.0 | 50.4 | 67.3 | 82.1 | 29.8 | 43.1 | 70.2 | 17.6 | 29.7 | 65.1 | 8.0 | 17.7 | 66.6 | 39.1 | 56.4 | 79.0 | 20.9 | 32.8 | 67.4 | 11.5 | 19.6 | 60.4 |
| TAPIR [9] | 17.6 | 30.7 | 61.0 | 17.8 | 29.1 | 56.0 | 12.7 | 19.3 | 62.8 | 10.5 | 15.5 | 65.8 | 9.7 | 13.8 | 75.9 | 21.3 | 30.3 | 68.1 | 9.14 | 14.6 | 61.3 | 3.7 | 6.5 | 58.7 |
| BootsTAP [8] | 20.5 | 32.9 | 67.8 | 30.7 | 42.8 | 71.0 | 17.4 | 25.3 | 65.4 | 12.2 | 17.1 | 66.5 | 12.4 | 16.8 | 76.1 | 30.3 | 39.3 | 76.9 | 11.4 | 17.7 | 62.8 | 3.7 | 6.5 | 58.6 |
| TAPTR [21] | 29.3 | 38.3 | 74.8 | 27.5 | 33.7 | 87.0 | 20.0 | 28.4 | 73.8 | 15.6 | 21.5 | 68.2 | 7.1 | 12.3 | 72.6 | 27.4 | 34.7 | 80.2 | 14.5 | 21.2 | 68.7 | 5.7 | 9.5 | 62.0 |
| TAPNext [? ] | 30.0 | 41.5 | 75.7 | 45.9 | 57.0 | 75.2 | 30.6 | 38.9 | 75.2 | 17.6 | 23.9 | 71.7 | 7.8 | 14.4 | 74.3 | 37.5 | 43.7 | 72.1 | **24.0** | **34.9** | **72.7** | **22.4** | **33.7** | **82.5** |
| SpatialTracker [34] | 27.2 | **38.0** | 71.6 | 45.8 | **60.1** | 81.4 | 26.0 | 37.4 | 70.5 | 17.3 | 26.8 | 69.1 | **9.9** | **17.2** | 71.7 | 40.4 | **53.3** | 82.8 | 16.3 | **25.2** | 66.0 | 8.7 | **15.3** | 59.8 |
| SceneTracker [31] | 21.5 | 21.3 | 52.0 | 32.2 | 59.3 | 61.0 | 14.8 | 37.8 | 49.2 | 8.9 | 23.9 | 43.5 | 2.9 | 16.3 | 31.2 | 24.6 | 51.9 | 53.5 | 11.1 | 25.8 | 47.6 | 6.8 | 14.9 | 48.2 |
| DELTA (2D) [23] | 27.1 | 35.2 | 70.8 | 48.2 | 59.6 | 84.0 | 26.5 | 35.2 | 70.0 | 14.9 | 21.5 | 68.1 | 8.1 | 12.5 | 72.3 | 40.3 | 50.0 | 81.6 | 14.6 | 21.7 | 65.4 | 7.4 | 11.5 | 58.8 |
| DELTA (3D) [23] | **29.1** | 37.2 | **74.1** | **48.7** | 58.8 | **85.1** | **28.8** | **38.7** | **71.3** | **19.0** | **27.1** | **71.5** | 9.2 | 15.7 | 69.4 | **43.6** | 53.0 | **85.3** | **17.1** | 25.1 | **68.7** | 8.4 | 13.5 | **60.8** |

Table 4: **ITTO vs. TAP-Vid.** SOTA 2D and 3D tracker performance on TAP-Vid and ITTO benchmarks.

| | Model | | | | | |
| | CoTracker3 | | | Delta (3D) | | |
| | AJ | $\delta$ | OA | AJ | $\delta$ | OA |
|---|---|---|---|---|---|---|
| ITTO | 35.5 | 47.1 | 77.7 | 29.1 | 37.2 | 74.1 |
| TAP-Vid Davis [7] | 64.4 | 76.9 | 91.2 | 64.2 | 77.3 | 87.8 |
| TAP-Vid Kinetics [7] | 54.7 | 67.8 | 87.4 | 50.3 | 63.5 | 83.2 |

Table 5: **Performance along occlusion rate tiers.** Performance of all SOTA models drops for tracks that have higher number of occluded frames.

| | Track Occlusion Rate | | | | | | | | |
| | (0, 24] | | | (24, 72] | | | (72, 100] | | |
| | AJ | $\delta$ | OA | AJ | $\delta$ | OA | AJ | $\delta$ | OA |
|---|---|---|---|---|---|---|---|---|---|
| CoTracker3 [17] | 43.9 | 50.6 | 76.7 | 30.2 | 41.8 | 73.5 | 24.7 | 39.4 | 83.5 |
| LocoTrack [3] | 42.6 | 50.3 | 78.5 | 22.0 | 40.3 | 62.8 | 12.5 | 35.0 | 65.0 |
| SpatialTracker [34] | 34.8 | 41.4 | 71.7 | 22.0 | 33.8 | 67.1 | 17.2 | 30.7 | 74.4 |

## 4.2 Model performance on ITTO

Table 3 reports the performance of current state-of-the-art 2D and 3D tracking models on ITTO. For a fair comparison, we report the best-performing variant of each model, using their publicly released checkpoints and inference code. We resize input videos to native model resolution before resizing the back to standard $[256 \times 256]$ for evaluation. We report both overall performance and performance delineated by axes of motion complexity. We run experiments on an NVIDIA A100. As methods are originally built on varying operating systems and software versions, we first replicate each model's results on existing benchmarks to verify that our reproductions match reported metrics.

From Table 3 we see that the highest reported performance on ITTO is significantly worse than other benchmarks and performance of all models degrades significantly along each axis of motion difficulty. LocoTrack and CoTracker3 perform best overall while there is a significant drop in other models. Notably, the relative ranking of models differs from their reported performance on TAP-Vid. For example, DELTA reports within 2% performance of CoTracker3, but drops by 6% on ITTO. Similarly, BootsTAP reports a 1.8% increase in AJ relative to CoTracker3, but drops by 15%. As most models typically report metrics within 5 percentage points of one another, this divergence suggests that ITTO introduces challenges not captured by existing benchmarks and that current models may be overfitting to those datasets and their characteristics. Performance also declines consistently with increased motion and reappearance frequency. For instance, on high-motion tracks, CoTracker3's AJ drops from 55.9 to 13.7 and $\delta$ from 68.4 to 22.1. Tracks with three or more reappearances show similar degradation, with AJ falling from 50.1 to 13.7 and $\delta$ from 60.5 to 21.5. We explore these findings in more detail below and provide qualitative examples from CoTracker3 on ITTO in Figure 4.2.

**Comparison to TAP-Vid.** Table 4 compares the best performing 2D and 3D trackers, CoTracker3 [17] and Delta [23] respectively, on ITTO and TAP-Vid. We find that the standard tracking metrics, Average Jaccard (AJ), points within delta ($\delta$), and occlusion accuracy (OA) are significantly worse on ITTO compared to TAP-Vid Davis and TAP-Vid Kinetics, suggesting that the motion complexities of the videos in ITTO make it an intrinsically more challenging benchmark.

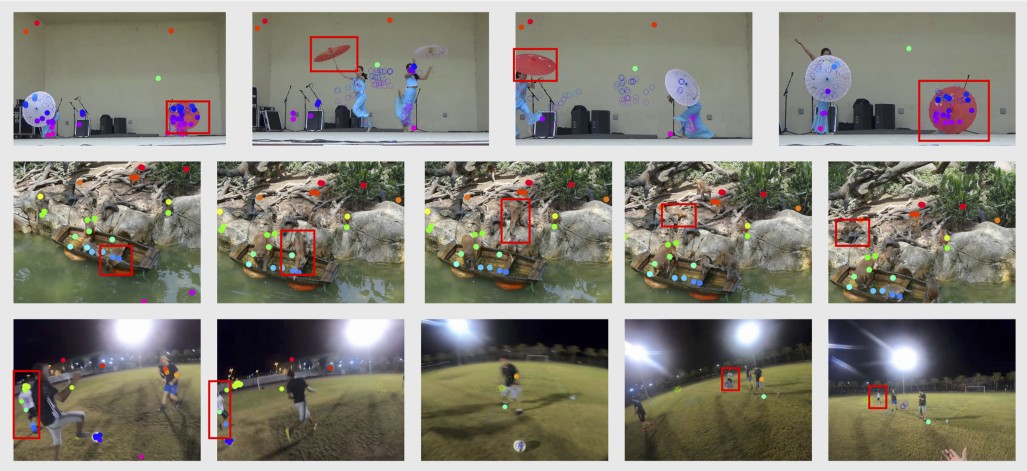

Figure 2: **CoTracker3 on ITTO.** Points tracked by color & red boxes added for illustration. Top: model biases red umbrella tracks to the right panel and fails when the girls switch positions. Middle: model struggles to track individual monkeys, which have repeated features. Bottom: model fails under camera-to-object distance changes. Background points are tracked with near-perfect accuracy.

**Tracking difficulties on complex motions.** The metrics in Table 3 reveal that models exhibit near-complete collapse on tracks with high motion. As frame-to-frame track motion increases, performance drops sharply and uniformly: for CoTracker3, Average Jaccard (AJ) falls from 55.9 to 13.7, and $\delta$ accuracy drops from 68.4 to 22.1 when track motion exceeds 5% of the frame diagonal. This suggests an inherent bias in models such as CoTracker3 that initialize stationary tracks. Performance is also worse for models with more nuanced track initialization based on feature matching [9, 3]: LocoTrack's AJ drops from 55.0 to 11.2, and $\delta$ drops from 65.4 to 16.6. We suspect this is because many ITTO videos have repeated objects and textures that switch positions several times, making it insufficient and often detrimental to rely on appearance-based track initialization biased towards specific motion or appearance patterns. Furthermore, this suggests an over-reliance on image feature correlations. 3D models also struggle on fast moving tracks; for SpatialTracker, AJ on fast-moving tracks is 9.9 (2.8% worse than CoTracker3) and $\delta$ is 17.2 (4.9% worse than CoTracker3). As our videos contain many similar objects, additional spatial consistency priors such as SpatialTracker's as-rigid-as-possible constraint and learned rigidity embedding might be insufficient to preserve point-to-object binding, as models might misplace tracks during fast motions.

We similarly see significant performance decline on tracks that undergo high rates of reappearance, suggesting that current trackers are unable to recover tracks after they undergo occlusion, especially repeated occlusions. For CoTracker3, AJ drops from 50.1 to 13.7 and $\delta$ drops from 60.5 to 21.5 on tracks that reappear more than 3 times in the video. 3D models also struggle on high-reappearance tasks. SpatialTracker has an AJ of 8.4 and $\delta$ of 13.5, suggesting that its 3D constraints also hurt with handling occlusions. Crucially, these failures cannot be explained by limited model context. The average length of an occlusion segment for tracks with 1 or fewer reappearances is 50.86 frames, for tracks between 2 and 3 reappearances is 23.15 frames, and for tracks with 3 or more reappearances is 12.46 frames. This indicates that models fail even on short occlusions well within typical input windows. The issue is not temporal reach but rather a fundamental inability to recover identity after occlusion, highlighting a major design gap in current architectures.

**Catastrophic tracking failure.** How well can models track through the motion difficulties of ITTO videos? We investigate this in Fig 4.2 by looking at track failure with respect to frame index (time). We define track failure rate as the percentage of tracks that are 2, 4, and 6 pixels outside of ground truth (model outputs are rescaled to $[256 \times 256]$ resolution for fairness). We find that tracks quickly degrade for frames outside of models' native resolution windows and remain broken for the rest of the video. Most models rely on a sliding window and stationary track initialization to produce tracks over an entire video, which is susceptible to error drift, causing a model to "forget" query point features at later video frames. Our experiments suggest that this error drift is irrecoverable: if a track is occluded beyond the model's input resolution, the model fails to recover the track upon reappearance. One exception to this behavior is LocoTrack, which has a more sophisticated track

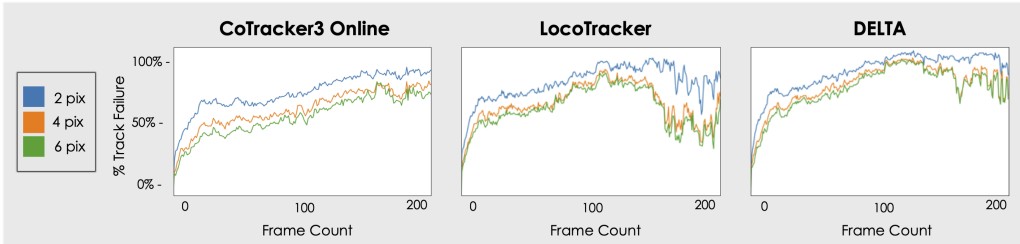

Figure 3: **Catastrophic Track Failure.** Track failure rates over time for 2, 4, and 6 pixel thresholds. CoTracker3 and DELTA suffer rapid and irrecoverable degradation once tracks leave their native resolution window. LocoTrack is more resilient, likely due to its global track initialization stage.

initialization step based on correlations in all frames of video, which could explain its ability to recover from errors. To deploy trackers in real-world settings, our findings highlight the need for either a track re-initialization step or a form of memory to supplement the auto-regressive nature of current models.

**Occlusion rate and occlusion accuracy.** There are many other ways to delineate motion complexity, but we found track motion and reappearance frequency to be the most informative in shedding insights into model behavior and limitations. Table 5 reports model performance categorized by the overall occlusion rate of each track. Although model performance degrades (for CoTracker3, AJ drops from 43.9 to 24.7 and $\delta$ drops from 50.6 to 39.4), this degradation is not as significant as it is for reappearance frequency. Interestingly, occlusion accuracy does not degrade with the degree of occlusion in a track: for CoTracker3, the occlusion accuracy is worse for tracks with occlusion rates of 0-73% than for those with 72-100% occlusion. We note similar behavior for TAPTR and SpatialTracker, as well as for other models which we omit for the sake of brevity. This suggests that occlusion accuracy as a standalone metric is insufficient, and that focus should be placed on track reappearances, which better capture a model's ability to track through occlusions.

## 5 Conclusion

In this paper we introduce ITTO, a challenging benchmark suite for evaluating tracking models under real-world motion complexity. ITTO contains key aspects of real-world motion absent from current benchmarks, and we report a much lower performance by state-of-the-art trackers than currently reported benchmarks. We propose ITTO as a holistic testing protocol for identifying strengths and weaknesses of new tracking models, and we introduce new axes of motion complexity, frame-to-frame displacement and reappearance count, to better understand model limitations. We report degrading performance for all models along each axis and an almost complete collapse under extreme difficulty. This suggests that current models are ill-equipped to handle extreme motion and reappearance, and are instead overfitting to current benchmarks. We attribute this limitation to biases introduced by stationary and feature-based track initialization, which are less effective on videos with complex motions and repeated features. We also observe that existing approaches fail to maintain trajectories beyond their built-in context windows and suffer from severe forgetting. Because most benchmarks place little emphasis on handling re-appearance, current methods are not designed to tackle this challenge. This gap highlights the need to integrate novel formulations such as memory mechanisms, coarse-to-fine approaches, or object pointers into current model architectures.

ITTO has several limitations and exciting avenues for future work. While our videos are real-world, extending to 3D could surface motion challenges also missing in 3D tracking benchmarks. While our data spans varied environments, many domains remain uncovered; we release our annotation pipeline to support community-driven, domain-specific extensions. Our reliance on human annotations limits precision to the pixel level, which may limit applicability in high-precision contexts like surgery. We also omit featureless regions (e.g., sky, water), which remain key failure modes for trackers. ITTO is not designed to "break" models, but to uncover essential real-world tracking challenges overlooked by current benchmarks. We offer it as a new benchmarking protocol for advancing tracking and guiding future model development.

## Acknowledgments and Disclosure of Funding

We would like to thank Damiano Marsili, Aadarsh Sahoo, and Ziqi Ma for valuable feedback and discussion. Ilona is supported by the Caltech EAS Chair Scholarship and the National Science Foundation Graduate Research Fellowship Program under Grant No. 2139433. This project was funded by the Caltech-BP program.

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

# 6 Appendix

We provide formal definitions and detailed analyses of motion and occlusion complexity that are addressed by the ITTO benchmark. We next describe our annotation pipeline and point query generation process. Finally, we report SOTA tracking model performance on ITTO along additional axes of motion complexity, and provide extended qualitative examples illustrating model behavior on the most challenging motion scenarios in ITTO.

## 6.1 Motion complexity

In this section we provide definitions for the metrics of motion and occlusion complexity that we use in both in our analyses of existing benchmark datasets compared to ITTO, as well as in our investigation of SOTA model performance on ITTO.

**Track motion.** We define two displacement-based motion metrics for each trajectory: frame-to-start motion ($\mathrm{disp}_{\mathrm{start}}$) and frame-to-frame motion ($\mathrm{disp}_{\mathrm{rel}}$). For each visible frame $i \in [1, T]$:

$$\mathrm{disp}_{\mathrm{start}} = \sqrt{(x_i - x_0)^2 + (y_i - y_0)^2}, \quad \mathrm{disp}_{\mathrm{rel}} = \sqrt{(x_i - x_{i-1})^2 + (y_i - y_{i-1})^2}$$

Motion values are computed in both pixel coordinates and normalized units (as a fraction of the frame diagonal). We characterize overall track motion using the average of these displacement values across time. For analysis, tracks are grouped by their average frame-to-frame motion into four intervals: $[0, 0.5)\%$, $[0.5, 1.5)\%$, $[1.5, 5)\%$, and $[5, 100)\%$ of the frame diagonal.

**Occlusion rate.** We define the occlusion rate following standard definitions as the ratio of occluded points to total points:

$$\text{occlusion rate} = \frac{\text{\# occluded points}}{\text{total points}}$$

**Bounding-box occlusion rate (BOR).** BOR reflects the occlusion degree for an object's bounding box. Following the object segmentation literature [25, 35] we define as the IoU between object mask $B_i$ and all other masks $B_j$:

$$\mathrm{BOR} = \frac{\left| \bigcup_{1 \leq i < j \leq N} \mathbf{B}_i \cap \mathbf{B}_j \right|}{\left| \bigcup_{1 \leq i \leq N} \mathbf{B}_i \right|},$$

We also calculate a frame-level per-object BOR, which reflects the amount that each individual object mask is occluded:

$$\mathrm{BOR}_i = \frac{\left| \bigcup_{1 \leq j \leq N, i \neq j} \mathbf{B}_i \cap \mathbf{B}_j \right|}{|\mathbf{B}_i|}$$

### 6.1.1 Detailed motion analyses

We provide a detailed breakdown of track motion across existing benchmarks and ITTO. Table 6 shows the percentage of tracks falling into motion intervals based on average frame-to-frame displacement, measured as a percentage of the frame diagonal. ITTO contains a significantly higher proportion of fast-moving tracks: 43.8% fall in the 1.5–5.0% range and 10.1% exceed 5.0% frame-to-frame motion. In contrast, TAP-Vid Davis has only 17.0% in the 1.5–5.0% range and 0.7% above 5.0% motion, while TAP-Vid Kinetics has just 5.5% and 0.3% in each respective category. This disparity reflects the low-motion bias in currently used benchmarks, where a dominant foreground object typically moves with the camera, resulting in largely stationary tracks.

The increased motion complexity of ITTO is further demonstrated by the track displacement statistics in Table 7. We report both frame-to-start and frame-to-frame motion, measured in pixels (using native resolution) and in normalized coordinates (relative to the frame diagonal). The average normalized frame-to-start motion in ITTO is 23.0, substantially higher than 13.9 in TAP-Vid Davis. For frame-to-frame motion, ITTO averages 3.6, while TAP-Vid Davis reaches only 1.2. All other benchmarks

|  | [0,0.5) | [0.5, 1.5) | [1.5, 5) | [5, 100] |
|---|---|---|---|---|
| TAP-Vid Davis | 27.4 | 54.9 | 17.0 | 0.7 |
| TAP-Vid Kinetics | 64.2 | 30.0 | 5.5 | 0.3 |
| TAP-Vid RGB Stacking | 62.6 | 28.9 | 8.3 | 0.2 |
| DynamicReplica | 96.0 | 3.8 | 0.2 | 0.0 |
| ITTO | 18.3 | 27.8 | 43.8 | 10.1 |

Table 6: **Motion breakdown of existing benchmarks and ITTO**. We report the percentage of tracks falling into four frame-to-frame motion intervals, expressed as a percentage of motion relative to the length of the frame diagonal: $[0, 0.5)$, $[0.5, 1.5)$, $[1.5, 5)$, and $[5, 100]$. Existing benchmarks are dominated by low-motion tracks (under 1.5%), while ITTO includes a significantly higher proportion of mid- and high-motion tracks (above 1.5%), better reflecting real-world tracking challenges.

|  | frame-to-start | | | | frame-to-frame | | | |
|---|---|---|---|---|---|---|---|---|
|  | normalized | | pixels | | normalized | | pixels | |
|  | mean | std | mean | std | mean | std | mean | std |
| TAP-Vid Davis | 13.9 | 13.6 | 35.6 | 35.0 | 1.2 | 2.0 | 3.0 | 5.0 |
| TAP-Vid Kinetics | 11.1 | 15.0 | 28.4 | 38.4 | 0.5 | 1.5 | 1.3 | 3.9 |
| TAP-Vid RGB Stacking | 10.1 | 15.6 | 25.8 | 39.8 | 0.4 | 1.5 | 1.0 | 3.8 |
| DynamicReplica | 8.1 | 10.8 | 65.3 | 77.3 | 0.2 | 0.6 | 1.6 | 5.1 |
| ITTO | 23.0 | 20.1 | 311.1 | 330.1 | 3.6 | 6.5 | 46.7 | 92.7 |

Table 7: **Motion complexity of existing benchmarks and ITTO**. We report mean and standard deviation of two motion metrics for each dataset: frame-to-start and frame-to-frame displacement. Values are shown in both normalized coordinates (relative to the frame diagonal) and in pixels. ITTO exhibits substantially higher motion across both metrics, reflecting its greater motion complexity relative to existing tracking benchmarks.

report even lower values, with frame-to-frame motion typically below 0.5, meaning most tracks move less than 0.5% of the frame per timestep. This indicates a strong stationary bias in existing datasets, where tracking is often reduced to modeling small, smooth deformations rather than dynamic and multi-object motions that ITTO contains.

## 6.2 Multi-object motion tracking

As ITTO contains videos with complex multi-object motions, another interesting question to explore is how robust current tracking models are to these interactions. To address this, we propose Pairwise Distance Variance (PDV) as a complementary metric that captures the spatial coherence of tracks belonging to the same object mask throughout the duration of the video. We define PDV for the points $p_i(t) \in \mathbb{R}^2$ belonging to the same object as follows: for each pair of points $(i, j)$ at each time $t$, define the pairwise distance $d_{ij}(t) = ||p_i(t) - p_j(t)||_2$. We define the pairwise $PDV_{ij} = \frac{\sigma_{ij}^2}{\bar{d}_{ij}^2}$ where $\sigma_{ij}^2 = \frac{1}{T-1} \sum_{t=1}^{T} (d_{ij}(t) - \bar{d}_{ij})^2$ is the variance of distances for each pair and $\bar{d}_{ij} = \frac{1}{T} \sum_{t=1}^{T} d_{ij}(t)$ is the mean distance for each pair. The overall $PDV$ is the mean $PDV_{ij}$ over all track pairs belonging to the same object mask (we note that we do not calculate a pairwise distance at frames where at least one track is occluded).

The intuition behind the $PDV$ is that if an object moves rigidly and in parallel to the camera, points on the object will maintain fixed spatial relationships so we expect the $PDV$ to be close to 0. In a deformable object, these pairwise distances change as the object stretches, compresses, bends, or undergoes occlusion, so we expect the $PDV$ to be of higher value. The square root of the $PDV$ thus provides us with the relative distance variation of tracks belonging to a given object. We define object motion to be simple if its tracks have a $PDV < 0.05$ (corresponding to $< 22\%$ relative deformation), and motion to be complex if its tracks have a PDV 0.05. We find that all track metrics are worse on complex object motions: for CoTracker3 (the most performant model), AJ goes from 45.2 to 29.5, $\delta$

|  | Track PDV | | | | | |
| | < 0.05 | | | ≥ 0.05 | | |
| | AJ | δ | OA | AJ | δ | OA |
| CoTracker3 [17] | 45.2 | 60.4 | 81.8 | 29.5 | 42.6 | 73.0 |
| LocoTrack [3] | 45.0 | 68.2 | 79.7 | 25.6 | 44.8 | 68.5 |
| DELTA 3D [23] | 38.8 | 55.3 | 78.2 | 22.8 | 32.8 | 65.7 |

Table 8: **Performance along PDV rate tiers.** Performance of all SOTA models drops for tracks that have higher PDV.

goes from 60.4 to 42.6, and OA goes from 81.8 to 73.0. We report the best-performing PDV metrics in Table 8. As many tracking architectures rely on attention steps between track features and iterative deformations of stationary tracks, this finding makes sense: the more complex the track motions are, the more difficult it is for the model to establish spatial feature correlations and calculate more complex track deformations.

## 6.3 Annotation tools

We employ a two-stage annotation pipeline to ensure high-quality tracking labels. In the first stage, we collect coarse annotations using Amazon Mechanical Turk (MTurk). Annotators are shown a reference frame and a query frame side by side. The reference frame includes a yellow arrow indicating the point to track, while the query frame comes from another timestep in the video. If the point is visible in the query frame, annotators place a visible marker at the proper location; if occluded, they select "mark as hidden." Annotators scroll through all frames in the video, labeling visibility on a per-frame basis. For videos with frame 0 queries, frames are presented in chronological order; for videos with nonzero query frames, we present frames from the query frame forward, then backward to frame 0, ordering along temporal proximity.

In the second stage, we refine these coarse annotations using a custom video annotation tool and a team of trained UpWork annotators. We initialize the tool with MTurk annotations and have annotators correct point positions and visibility labels by playing through the video, changing visibility flags and moving annotated points to the correct locations with the arrow keys. The video tool includes keyboard shortcuts for efficiency and a frame-stepping interface that supports 1- and 5-frame increments in both directions, which is particularly useful in fast-motion or occluded sequences. Annotators are instructed to play through each video multiple times to ensure spatiotemporal consistency. A reference image is also provided at the bottom of each task screen to mitigate drift, especially in longer or more complex sequences. Figure 4 illustrates both stages of the pipeline.

### 6.3.1 Pseudomasks for Algorithmic Point Querying

For the MOSE and LVOS subsets of our dataset, we leverage available ground-truth segmentation masks to algorithmically sample query points directly from object regions, as described in the main text. For the egocentric portion, sourced from Ego4D, no such ground-truth masks exist. To ensure object-centric point sampling, we generate pseudomasks using a two-step pipeline. First, we query Molmo with object-level semantic prompts (e.g., "tree," "hat," "hand") specific to the query frame. We prompt Molmo with: *"Point to the {object}, and give me three coordinates"* for positive samples, and *"Point to coordinates that are not a {object}, and give me two coordinates"* for negative samples. These coordinates are then passed to SAM2, which produces segmentation masks from the provided positive and negative points. We found that including negative prompts significantly improved mask quality by reducing spurious activations. Figure 5 illustrates several examples of the resulting pseudomasks for videos in our dataset.

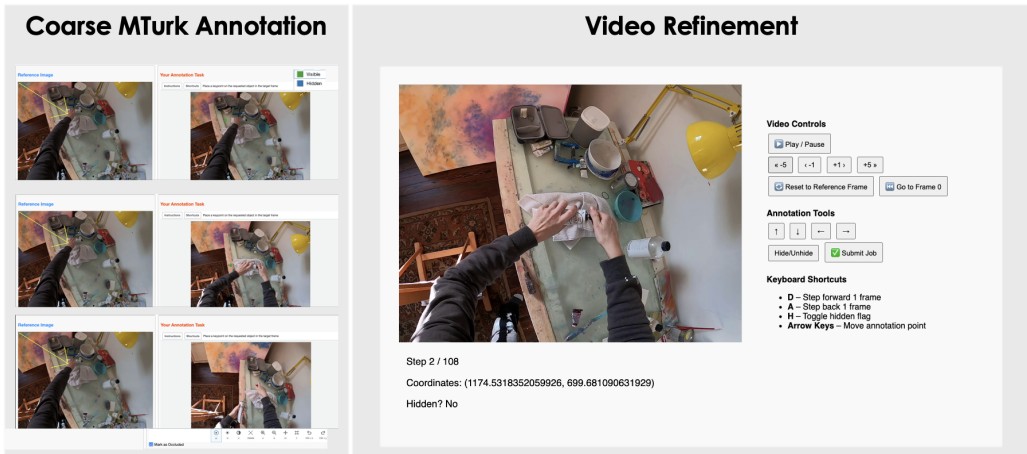

Figure 4: **Annotation Pipeline**. Left hand side shows Amazon Mechanical Turk setup. MTurk workers are given a reference image and a query image side-by-side, and place visible keypoints on query images for which the point is visible. If the point is occluded, they select "label as hidden'" for that frame. Right-hand side shows the video refinement tool. Annotators play through the video and correct the point location and visibility flag through the arrows on the control panel. Alternatively they can also use keyboard controls to play through the video and refine the tracks. Annotators play through the video and correct frames as many times as they need until the track is spatiotemporally consistent.

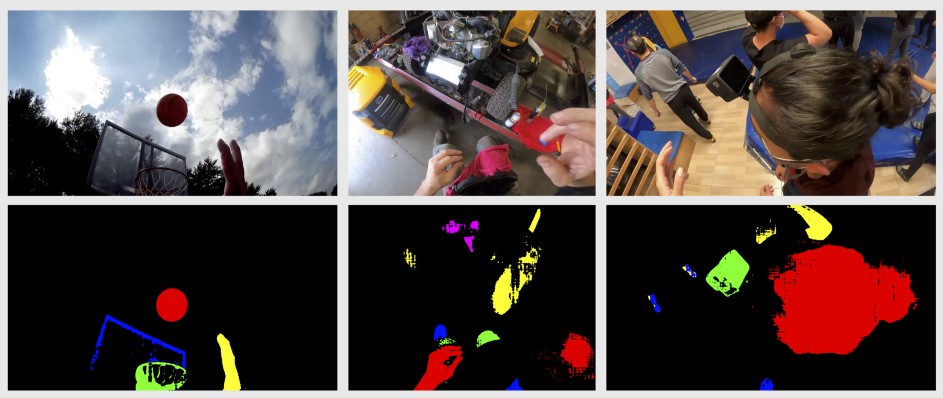

Figure 5: **Pseudomasks**. We generate pseudomasks that we generate by first prompting Molmo to generate three positive and two negative coordinates for input semantic concepts. We send these labels into SAM2, which generates pseudomasks. For the leftmost video, we prompt with "ball", "frame", "net", and "hand"; for the middle video we prompt with "engine", "tire", "hand" "foot", and "pant"; and for the rightmost video we prompt with "head", "GoPro", "hand", and "forearm".

## 6.4 Model performance on ITTO

In Table 9 we report SOTA model performance on additional motion complexity metrics. We report performance along increasing occlusion rate for all SOTA methods, in addition to CoTracker3, LocoTrack, and SpatialTracker which we report in the main text (Table 5). Consistent with trends observed in Table 5 of the main text, we find that performance degrades as occlusion increases. Among 2D methods, CoTracker3 online performs best: AJ degrades from 43.9 to 24.7 and $\delta$ degrades from 50.6 to 39.4. For 3D methods, DELTA 3D performs best under low occlusion, but SpatialTracker does better in high-occlusion settings, outperforming DELTA by 8.2 points in AJ and 15 points in $\delta$. For all models, accuracy does not correlate with model performance, and in fact occlusion accuracy is higher for the tracks with high occlusions, aligning with our qualitative finding that models frequently default to predicting tracks as occluded when motion becomes complex. This suggests that occlusion

Table 9: **Performance of SOTA tracking models on ITTO for additional motion complexity buckets**. We report performance along motion complexity tiers of track motion and number of visible consecutive frame segments per track, and see further significant degradation for high track motion and visible track segment counts. Horizontal line separates 2D methods (top) from 3D methods (bottom).

| | Track Occlusion Rate | | | | | | | | | Query type | | | | | | | | |
| | (0, 24] | | | (24, 72] | | | (72, 100] | | | Gradient | | | Random | | | Background | | |
| | AJ | δ | OA | AJ | δ | OA | AJ | δ | OA | AJ | δ | OA | AJ | δ | OA | AJ | δ | OA |
|---|---|---|---|---|---|---|---|---|---|---|---|---|---|---|---|---|---|---|
| CoTracker3 online [17] | **43.9** | **50.6** | **76.7** | **30.2** | **41.8** | 73.5 | **24.7** | **39.4** | 83.5 | **33.2** | **45.8** | **75.6** | **34.3** | **48.3** | **80.3** | **39.1** | **49.7** | **79.7** |
| CoTracker3 offline [17] | 42.4 | 48.9 | 73.4 | 29.2 | 39.4 | 74.4 | 21.7 | 30.0 | **83.6** | 30.8 | 43.0 | 74.2 | 32.7 | 43.4 | 80.1 | 38.1 | 48.5 | 79.4 |
| LocoTrack [3] | 42.6 | 50.3 | 78.5 | 22.0 | 40.3 | 62.8 | 12.5 | 35.0 | 65.0 | 14.6 | 25.1 | 62.4 | 13.4 | 22.4 | 63.3 | 11.2 | 21.5 | 61.4 |
| TAPIR [9] | 14.2 | 19.2 | 43.5 | 13.0 | 21.7 | 64.9 | 9.9 | 17.0 | 81.5 | 15.6 | 25.7 | 65.1 | 15.8 | 25.0 | 70.7 | 16.5 | 27.3 | 62.4 |
| BootsTAP [8] | 22.3 | 28.3 | 54.3 | 17.0 | 25.7 | 66.9 | 15.6 | 24.9 | 81.1 | 18.3 | 30.7 | 67.1 | 18.7 | 33.4 | 73.8 | 23.5 | 35.8 | 67.9 |
| TAPTR [21] | 11.3 | 16.6 | 53.9 | 15.0 | 20.3 | 60.2 | 21.3 | 29.2 | 84.2 | 8.9 | 14.0 | 48.1 | 14.0 | 21.9 | 58.3 | 6.9 | 9.8 | 29.2 |
| SpatialTracker [34] | 34.8 | 41.4 | **71.7** | 22.0 | **33.8** | 67.1 | **17.2** | **30.7** | **74.4** | 22.0 | **34.5** | 67.5 | 22.9 | **33.4** | 71.9 | 31.3 | **40.7** | 74.6 |
| SceneTracker [31] | 32.2 | **43.1** | 80.2 | 9.3 | 39.4 | 30.6 | 4.6 | 37.3 | 14.7 | 12.3 | 21.9 | 62.3 | 12.8 | 21.7 | 60.8 | 10.9 | 21.3 | 57.0 |
| DELTA (2D) [23] | 34.4 | 40.0 | 66.2 | 21.7 | 30.9 | 68.1 | 15.2 | 25.3 | 76.6 | 23.5 | 32.0 | **68.7** | 21.1 | 30.0 | 70.9 | 30.3 | 38.0 | 72.3 |
| DELTA (3D) [23] | **36.5** | 42.8 | 71.1 | **24.5** | 33.5 | **72.0** | 9.2 | 15.7 | 69.4 | **25.4** | 34.0 | 71.8 | **23.3** | 31.3 | **73.9** | **32.4** | 40.1 | **75.7** |

accuracy alone is an insufficient diagnostic and highlights the value of out motion-based complexity metrics of track displacement and reappearance frequency.

We also report model performance based on query type. Among 2D models, CoTracker3 online performs best, with an AJ of 33.2 for gradient-based queries, 34.3 for random-based queries, and 39.1 for background queries. Among 3D models, top performance is split between SpatialTracker and DELTA, with SpatialTracker having a higher δ metric for all query types. We find that models perform better on background points, which are often unmoving and is in line with our finding that these models exhibit a stationary bias. We find that there is a smaller difference in performance for gradient-based versus randomly sampled queries, suggesting that the difficulty is not in feature saliency but rather in tracking the objects themselves. These findings show that placing tracks on background points in videos can give a false signal for model performance, which we mitigate in our ITTO benchmark by ensuring that we do not over-sample background points.

## 6.5  Qualitative examples of model performance on ITTO

Figure 6 presents qualitative results from CoTracker3 Online on the three lowest-performing ITTO videos, ranked by Average Jaccard (AJ) scores of 2.3%, 4.0%, and 9.0% (top to bottom). In all three cases, the model rapidly predicts most tracks as occluded, often within the first few frames. In the top example, a bull video, tracks remain roughly localized on the object initially, but once the camera zooms in, predictions are quickly marked as occluded and eventually diverge entirely from the bull. In the middle video, which features two lizards, the model fails to differentiate between them, resulting in occlusion predictions and spatial collapse of tracks toward the midpoint between the two animals. In the bottom example, the model fails to consistently track points on a cat's fur, with predictions drifting first to the cat's head and eventually to the background (a towel). These failures occur well before the 16-frame input window length used by CoTracker3 online, suggesting that they are not caused by input resolution limits but by early feature-correlation collapse or failure in motion handling.

We observe similar failure patterns with LocoTrack and DELTA. As shown in Figure 8, LocoTrack shares the two most difficult videos with CoTracker3, indicating common weaknesses in handling fast object motion and occlusion. Its third most difficult video with respect to AJ is an egocentric basketball video, where it fails to maintain tracks on the basketball hoop during rapid camera movement and loses all tracks as the camera moves away from the scene. DELTA exhibits comparable issues (Figure 9). It also fails on the lizard video but additionally struggles with two egocentric clips. In the bottom row example, it cannot retain tracks on a moving hand. Although the hand's motion is relatively simple (the hand is essentially static relative to the camera frame), the model fails likely due to errors in the depth module or 3D motion representation, exacerbated by the rapid background motion.

We provide additional qualitative examples of the lowest-performing sequences for each model in Figures 10, 11, and 12. Across all models, we observe catastrophic failure modes and an inability to recover from early occlusion or incorrect track initialization. Models also struggle to distinguish between visually similar object regions, especially when tracking semantically distinct keypoints. These failures are further exacerbated by rapid object motion and abrupt camera changes, complexities contained in ITTO videos that contribute to its motion difficulty.

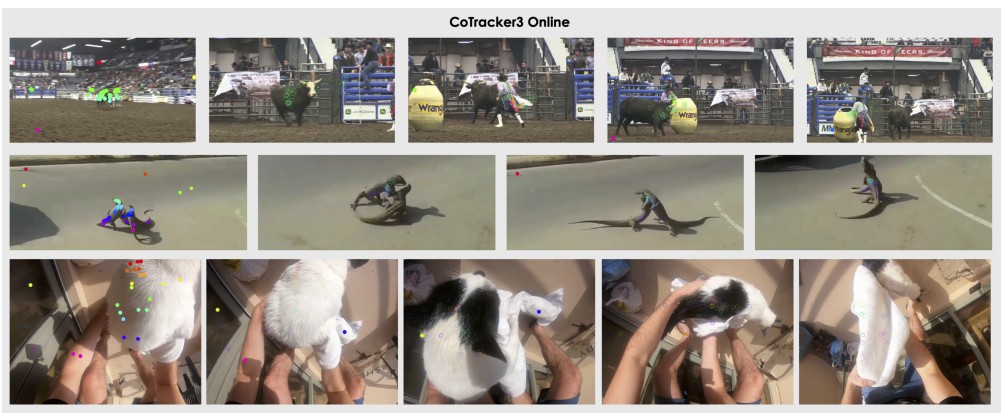

Figure 6: **Challenging ITTO videos for CoTracker3 online.** Visualization of the three videos with the lowest Average Jaccard (AJ) scores for CoTracker3 online. Track IDs are color-coded; empty circles indicate frames where tracks are predicted as occluded. Top: the model fails under rapid depth change as the camera zooms in. Middle: model cannot reliably distinguish between two visually similar reptiles. Bottom: model struggles to maintain tracks on the cat's fur, with predictions drifting to the head and towel. In all cases, tracks are prematurely marked as occluded.

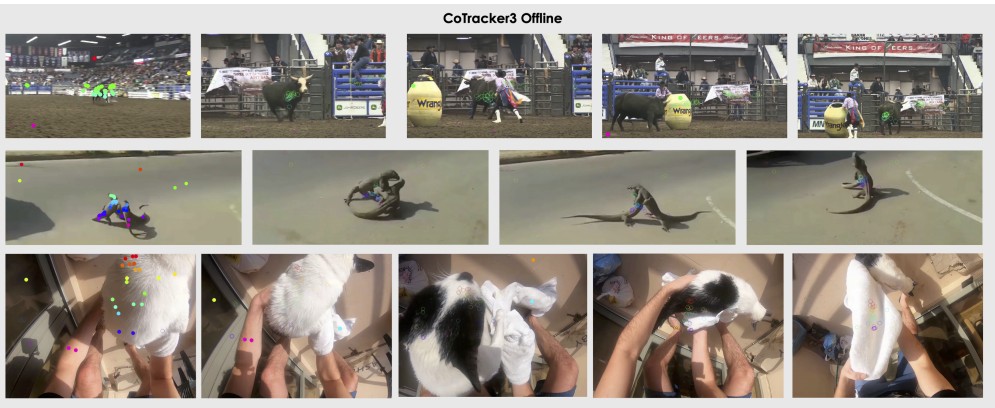

Figure 7: **Challenging ITTO videos for CoTracker3 offline.** Visualization of the three videos with the lowest Average Jaccard (AJ) scores for CoTracker3 offline. Offline model struggles on the same three videos as the online variant, however tracks remain on the bull in the third visualized frame, and drift off the bull in the fourth frame, before being placed back on the bull again in the last frame, the inverse of the online version. All tracks are incorrectly predicted as occluded in these frames.

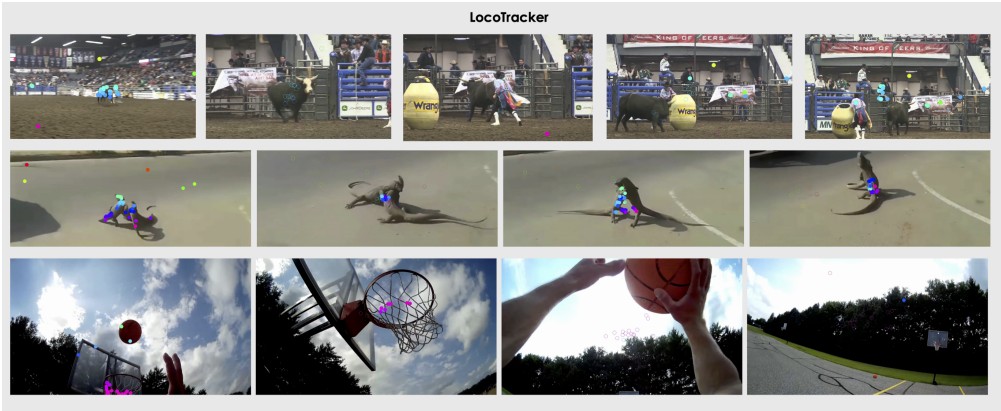

Figure 8: **Challenging ITTO videos for LocoTrack.** Visualization of the three lowest-performing videos for LocoTrack. Top: the model fails under rapid camera zoom, leading to incorrect occlusion predictions and misplaced tracks on background objects. Middle: the model is unable to differentiate between the two lizards, resulting in ambiguous and collapsed track predictions. Bottom: significant depth changes cause the model to lose all track points by the end of the basketball throw sequence.

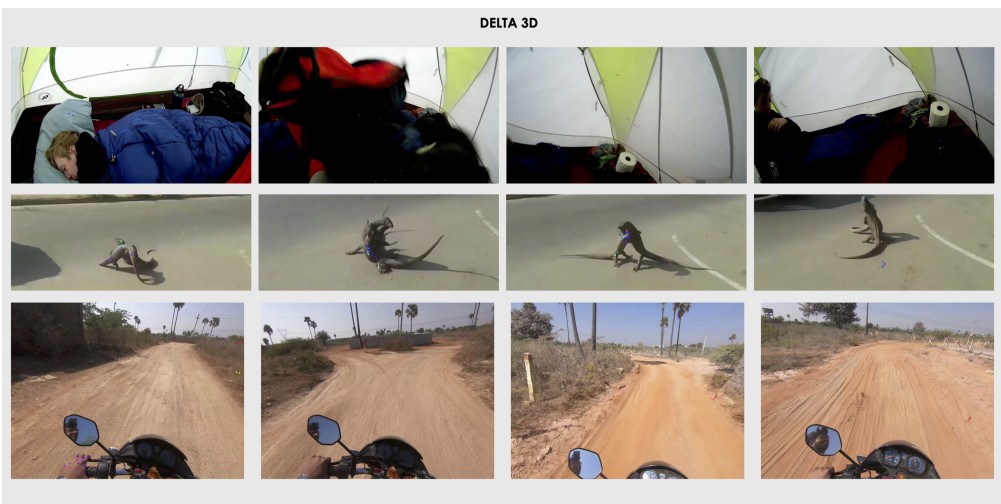

Figure 9: **Challenging ITTO videos for DELTA.** Visualization of the three lowest-performing videos for DELTA. Top: the model fails to track under rapid camera panning in a visually constrained environment (a tent), leading to early occlusion predictions. Middle: it cannot reliably differentiate between the two lizards, resulting in drift and eventual failure to track either. Bottom: the model is unable to maintain tracks on the hand, losing all keypoints early in the sequence, despite little-to-no motion relative to the camera frame.

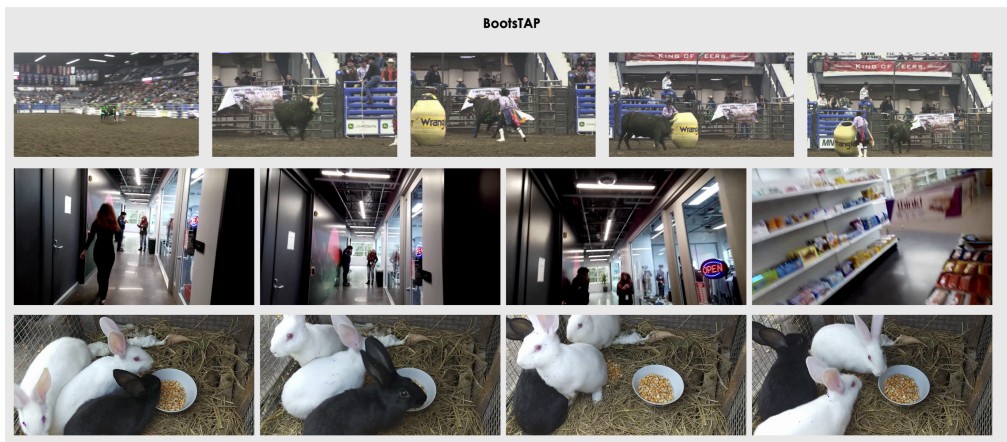

Figure 10: **Challenging ITTO videos for BootsTAP.** Visualization of the three lowest-performing videos for BootsTAP. Top: the model fails under rapid camera zoom, similar to other methods. Middle: while initial tracks on ceiling features are stable, the model is unable to recover later queried points, leading to degraded performance over time. Bottom: the model struggles to distinguish between two visually similar white rabbits, marking most tracks as occluded and concentrating predictions in the lower-left quadrant where coordinates were initially queried in frame 0.

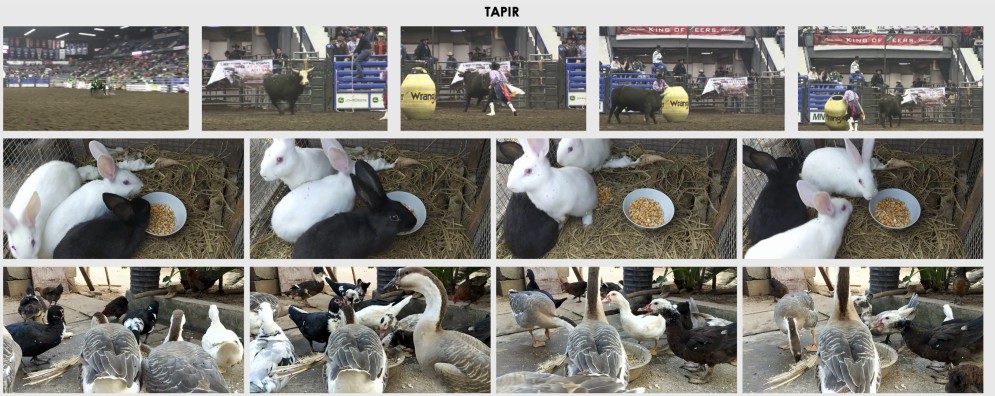

Figure 11: **Challenging ITTO videos for TAPIR.** Visualization of the three lowest-performing videos for TAPIR. Top: the model fails under rapid camera zoom, with tracks drifting away from the bull and remaining occluded for the remainder of the video. Middle: the model struggles to differentiate between two visually similar white rabbits, and confuses semantically distinct keypoints such as the nose and ears due to local feature similarity. Bottom: the model cannot distinguish between multiple geese with near-identical textures; tracks on the black geese are lost entirely, and most predictions are marked as occluded early in the sequence.

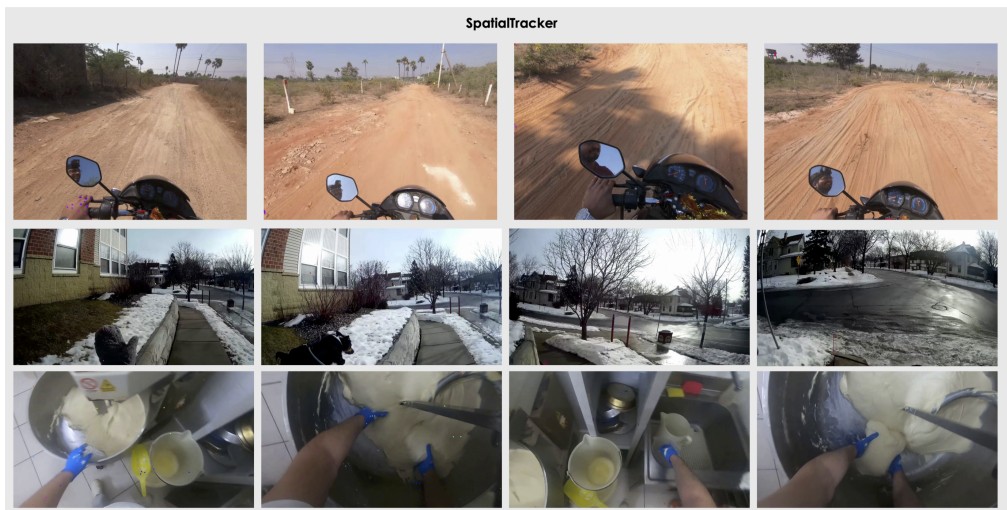

Figure 12: **Challenging ITTO videos for SpatialTracker.** Visualization of the three lowest-performing videos for SpatialTracker, all of which are egocentric sequences from ITTO, longer than 150 frames. Top: the model loses track of keypoints on a hand in the lower-left corner, despite it remaining relatively stationary with respect to the camera; partial occlusions lead to persistent occlusion predictions. Middle: the model fails when the headset camera rapidly changes viewing angle, resulting in loss of previously visible tracks. Bottom: the model loses all keypoints on the arm, pitcher, and mixing blade shortly after the video begins, with no recovery throughout the sequence.

