# OpenReview forum: "Is This Tracker On? A Benchmark Protocol for Dynamic Tracking"
_NeurIPS.cc/2025/Datasets_and_Benchmarks_Track — NeurIPS 2025 Datasets and Benchmarks Track poster_

### Official Review · Reviewer_fJSA · 2025-06-02

**Rating:** 4
**Confidence:** 5

**Summary:**

This paper introduces ITTO, a new benchmark suite for evaluating point tracking algorithms under realistic and challenging conditions. The dataset includes videos from existing sources as well as egocentric real-world recordings, annotated with a high-quality, multi-stage pipeline. ITTO is specifically designed to capture complexities often missing from current benchmarks, such as diverse object motion, occlusions, and real-world dynamics. The authors conduct a thorough evaluation of state-of-the-art tracking methods, breaking down performance across various dimensions of motion complexity. The results demonstrate that existing approaches struggle significantly, particularly with re-identification after occlusion, revealing key failure modes. Overall, ITTO provides a valuable testbed for diagnosing limitations in current methods and motivating the development of more robust point tracking solutions.

**Additional Feedback:**

I hope the authors will seriously consider these concerns. While I am inclined to give a positive recommendation, I believe a potential v2 version that addresses some of the above issues would significantly strengthen the impact and utility of this work.

**Dataset Code Accessibility:**

Yes

**Ethical Considerations:**

No, there are no or only very minor ethics concerns

**Final Justification:**

Most of my initial concerns have been addressed. This dataset captures complexities often missing from current benchmarks, such as diverse object motion, occlusions, and real-world dynamics, which makes it a valuable contribution to the point tracking task.
However, due to the lack of dense annotations and depth information, I cannot justify a higher score at this time. Therefore, I lean toward a borderline accept.

**Limitations Weaknesses:**

**Major Weaknesses:**

1. **Lack of a Training Set:** While ITTO serves as a valuable evaluation benchmark, it does not include a training set. Most existing methods are trained on synthetic datasets like Kubric, which contain relatively simple motion. As a result, although ITTO reveals that current methods fail under complex motion, it does not offer sufficient guidance for improvement due to the lack of diverse and realistic training data.

2. **Sparse Annotations:** The number of tracks per video is limited. Although TAP-Vid also uses sparse tracking, providing denser tracks would yield more information and allow for finer-grained evaluation.

3. **Sparse Query Sampling & Long Video Lengths:** In line 197, it is stated that queries are sampled at frame 0 for all videos, every 100 frames for LVOS, and at 1/4, 1/2, and 3/4 of the video length for Ego4D. This large temporal gap may miss key objects or events, especially those with brief appearances. Additionally, with videos averaging over 1000 frames, the length is excessive. Splitting videos into shorter clips and reducing the sampling interval could better exploit the information present in the videos.

4. **Comparison to More Data-Driven Methods:** The benchmark does not include comparisons with recent data-driven models such as TAPNext: Tracking Any Point as Next Token Prediction, which abandons explicit geometric priors like correlation. Such methods may generalize better across domains due to their data-centric design. Additionally, TAPTRv2, which integrates correlation into a transformer-based framework, deserves to be benchmarked as well. Including these models would provide a more comprehensive evaluation of current paradigms and their strengths and limitations under the challenging scenarios posed by ITTO.



**Minor Weaknesses:**

1. **Typographical Error:** In Table 2, there is a typo: "on bot hthe" should be corrected to "on both the".

2. **Lack of Formal Definitions:** The metrics “Track Motion” and “Reappearance Frequency” are somewhat intuitive, but a clear and concise mathematical formulation or example would significantly improve clarity. Although their general meanings can be inferred, formal definitions would help readers more precisely understand how these statistics are computed and interpreted.

3. **Missing Depth Information:** The benchmark currently does not incorporate depth information, which could be valuable for analyzing occlusion, motion complexity, and spatial reasoning. Including or aligning with depth cues may enhance the richness and diagnostic power of the benchmark.

**Strengths Contributions:**

1. The problem that ITTO aims to uncover is highly valuable. Many existing methods struggle with long-term tracking and tracking under fast motion due to their reliance on sliding window and local correlation mechanisms.

2. The paper provides detailed explanations and background information, giving us a clear understanding of how the proposed pipeline operates.

2. Extensive experiments are conducted across a wide range of baselines.

---

> ### Author Rebuttal · Authors · 2025-07-30
>
> We thank reviewer fJSA for the thoughtful and thorough review, and we are encouraged that they find that ITTO provides a valuable testbed for diagnosing the limitations in current methods and motivating the development of more robust point tracking solutions. There are several points raised by the review that we would like to address in the discussion below and that we will include in the camera-ready version of the paper:
>
> > 1. While ITTO serves as a valuable evaluation benchmark, it does not include a training set
>
> Our motivation for creating ITTO was to serve the community with a benchmark for evaluating and diagnosing the limitations of current tracking methods. The purpose of ITTO is to assess the zero-shot capability of state-of-the-art tracking models on real-world videos. Collecting such data is expensive, requiring several rounds of manual human annotation to ensure spatiotemporal consistency. ITTO exposes areas in existing evaluation datasets that are missing. For example in TapVID, the current de-facto tracking benchmark, there is very little data with significant track reappearance (Table 1), and in ITTO we show that models exhibit near collapse once a track reappears more than twice throughout the course of a video (Table 3). We believe that the community should focus on collecting training data that captures the complexity gaps that ITTO exposes.
>
> > 2. Providing denser tracks would yield more information and allow for finer-grained evaluation
>
> Indeed, ITTO does not annotate dense tracks, i.e. we don’t annotate all pixels with tracks. Instead we annotate a subset of pixels and focus on collecting high quality annotations through our two-stage pipeline. In Section 4, we show that the motion complexity and frame count of ITTO videos can help expose and isolate the limitations of existing methods, and we are excited about how it can inform future directions for model development.
>
> > 3. Queries are sampled at frame 0 for all videos, every 100 frames for LVOS, and at ¼, 1/2 , and ¾ of the video length for Ego4D. This large temporal gap may miss key objects or events, especially those with brief appearances.
>
> For the MOSE and LVOS portion of ITTO videos, all segmented objects appear in frame 0, which we include as a query frame across all sourced videos. Our algorithmic query point selection process ensures that every segmented object gets at least 1 query point, thus we do not miss any objects that contain ground-truth mask segmentations. We place more query points on object masks that exhibit higher frame-to-frame motion (however, as the reviewer notes, objects that briefly appear in later frames do not have ground truth masks and are thus omitted). This setup departs from current tracking benchmarks that rely on human annotators to pick which pixels to track, resulting in an inherent bias towards salient points. As a result, ITTO videos contain 4.25x as many tracked objects than current tracking benchmarks (see Table 2). In addition, while the average length of our tracks and videos is quite high (221.6 frames) our track annotations cover a range of appearance lengths including brief appearances: the length of each visible track segment ranges from a minimum of 1-10 frames to a maximum of 300 frames.
>
> > 4. With videos averaging over 1000 frames, the length is excessive. Splitting videos into shorter clips and reducing the sampling interval could better exploit the information present in the videos
>
> An important aspect of tracking is through long-range motions. Tracks with over 1000 frames assess whether models are able to track successfully through such temporally long motions without drifting, dropping, or confusing tracks. We are particularly excited by our catastrophic tracking failure finding in Section 4: we observe that existing approaches fail to maintain trajectories beyond their built-in context windows and suffer from severe forgetting. One exception to this is LocoTracker, which has a more sophisticated track initialization step based on correlations in all frames of a video, which could explain its ability to recover from errors. This suggests that better track initialization schemes could improve tracking through long range motions
>
> 5. Comparison to more data-driven methods such as TAPNext and TAPTRv2.
>
> This is a great point, and we would like to clarify that by the “TAPTR” model that we report in our paper, we are referring to the TAPTRv2 model, and that we will update the citation accordingly. Upon your suggestion, we also evaluate TAPNext (which was published four weeks before the NeurIPS deadline). TAPNext achieves an average jaccard (AJ) of 30.0, delta of 41.5, and occlusion accuracy (OA) of 75.7, which is on par with but not better than LocoTrack and CoTracker3 online, the most performant models (Table 3) – we will include these numbers in our camera ready. We also find similar trends to the ones we report in Section 4 along our proposed axes of difficulty: AJ drops from 43.9 to 7.8 and delta drops from 53.4 to 11.9 for tracks with higher reappearance counts; AJ drops from 45.9 to 7.8 and delta drops from 57.0 to 14.4 for tracks with higher track motion. While both of these methods formulate tracking as a detection problem, we hypothesize that they are less robust to frequent track reappearance or changing track appearance over longer video durations.
>
> 6. Lack of formal definitions for track motion and reappearance frequency
>
> We formally define track motion in Appendix A.1, and we define track reappearance count in Section 4.1 as the number of times a track transitions from occluded to unoccluded.
>
> 7. Missing depth information
>
> Our goal with ITTO is to evaluate 2D tracking and provide a diagnostics tool for state-of-the-art 2D and 3D trackers. It would be an exciting direction to extend to RGB-D videos, and our annotation pipeline could certainly be applied to such videos to yield high-quality annotations similar to ITTO! We will release our video annotation tool together with our evaluation code.

---

> > ### Comment · Reviewer_fJSA · 2025-08-04
> >
> > Thank you to the authors for the detailed rebuttal. Most of my initial concerns have been solved. I keep my positive score.

---

### Official Review · Reviewer_Ns1Q · 2025-06-25

**Rating:** 4
**Confidence:** 4

**Summary:**

This paper introduces a novel benchmark suite for dynamic point tracking, designed to address the lack of real-world motion complexity in existing benchmarks. ITTO sources videos from monocular and egocentric datasets, with high-quality annotations collected via a multi-stage pipeline. The benchmark captures complex non-rigid motions, occlusion patterns, and object diversity, which are absent in current datasets. Experiments show that state-of-the-art trackers (e.g., CoTracker3, LocoTrack) struggle significantly on ITTO, particularly in re-identifying points after occlusion, highlighting critical failure modes like stationary bias and memory limitations.

**Additional Feedback:**

None

**Dataset Code Accessibility:**

Yes

**Dataset Code Comments:**

The dataset, code, and annotation pipeline are openly available, enabling reproducibility and community adoption. The two-phase annotation protocol ensures high-quality labels .

**Ethical Comments:**

There are no ethical issues in this article.

**Ethical Considerations:**

No, there are no or only very minor ethics concerns

**Limitations Weaknesses:**

1. The dataset scale with 72 videos in ITTO is smaller than some existing benchmarks, which may limit generalizability across all real-world scenarios.

2. The paper introduces Track Motion as a key evaluation metric, partitioning tracks based on frame-to-frame displacement thresholds (e.g., <0.5%, 0.5–1.5%, >5.5% of frame diagonal) . While this metric captures individual track dynamics, it struggles to characterize the accuracy of multi-object motion tracking, where complex interactions (e.g., occlusions, non-rigid deformations, and co-dependent movements) dominate.

**Strengths Contributions:**

1. ITTO captures dynamic scenarios like object re-appearance and non-rigid motions, which existing benchmarks overlook.

2. The paper provides a systematic analysis of SOTA trackers in dynamic and complex scenes.

3. The findings highlight the need for memory mechanisms and dynamic modeling, guiding future research.

---

> ### Author Rebuttal · Authors · 2025-07-30
>
> We thank reviewer Ns1Q for the thorough review and are excited that they find that ITTO is a novel benchmark suite designed to address the lack of real-world motion complexity in existing benchmarks, and that our findings guide future tracking research. We would like to address two points below, and will include these discussions in the camera-ready version of the paper:
>
> 1. Dataset scale and quality trade-offs:
>
> We acknowledge the reviewer’s concern about dataset scale and appreciate the opportunity to clarify our rationale behind prioritizing high-quality complex data, which we will make sure to include in the camera-ready version of the paper. We believe that ITTO videos contain crucial aspects of motion complexity that are missing in existing benchmarks, and show that ITTO can be used to diagnose limitations of current tracking models.
>
> TAP-Vid Kinetics, despite its scale, contains predominantly simplistic motions: 88.1% of tracks are static compared to only 30.7% in ITTO (see Table 1). Track reappearances average just over 0.27 per video in Kinetics (meaning tracks rarely reappear once occluded) versus 5.86 in ITTO, where tracks repeatedly reappear and disappear from view, testing models’ ability to track through occlusions. Most tracking papers thus report results on TAP-Vid DAVIS (30 videos, 650 tracks) due to its more realistic and complex motions. ITTO substantially exceeds DAVIS across all complexity metrics, with 5x more tracked objects per video, 24% more object occlusions, 5.75x more frames, and 52% fewer static points across 2.4x more videos (Table 1). This positions ITTO as an important “next challenge” in current tracking evaluations.
>
> Furthermore, our systematic analysis in Section 4 demonstrates that specific motion challenges drive performance differences across models. We find that the performance of all state-of-the-art tracking models degrades significantly on ITTO videos for all metrics (AJ, delta, and OA). In Table 3, we demonstrate that specific motion complexities drive this performance degradation. The best reported Average Jaccard (AJ) drops from 55.9 to 13.7 and delta drops from 68.4 to 22.1 for ITTO tracks with higher track motion. The best reported AJ drops from 50.1 to 12.7 and delta drops from 60.5 to 21.5 for tracks with higher reappearance counts. These difficulty metrics are consistent across all 2D and 3D trackers; as videos get harder along our axes of motion difficulty, model performance drops. Crucially, all existing tracking benchmarks do not contain these aspects track motion, reappearance, or object counts. Thus we believe that ITTO is useful at gauging tracker performance and exposing its robustness to common aspects of real-world tracking difficulties. We will make sure to include a discussion of all of these points in the camera-ready version of the paper.
>
> While there is always room for dataset expansion given infinite resources, we believe that the diversity and complexity of motion in ITTO tracks is useful in isolating specific tracking failure modes and guiding future directions for model development (Section 4). Most importantly, ITTO contains complexity that is wholly missing from existing tracking benchmarks. In addition, ITTO spans diverse capture environments (ego/exocentric cameras, animal/human/driving scenes) that reflect real-world deployment scenarios for tracking applications in robotics, motion analysis, and multi-object settings.
>
> 2. Multi-object motion tracking:
>
> This is a great point: ITTO contains videos with complex multi-object motions so it is indeed interesting to explore how robust current tracking models are to these interactions. To address this, we propose Pairwise Distance Variance (PDV) as a complementary metric that captures the spatial coherence of tracks belonging to the same object mask throughout the duration of the video. We will include this metric and the proceeding discussion in the camera-ready version of the paper, as it sheds further insight into existing models’ limitations.
>
> We define PDV for the points ${p_{i}(t)} \in \mathbb{R}^2$ belonging to the same object as follows: for each pair of points $(i,j)$ at each time $t$, define the pairwise distance $ d_{ij}(t) = \lVert p_{i}(t) - p_{j}(t)\rVert_{2} $. We define the pairwise $PDV_{ij} = \frac{\sigma_{ij}^2}{d_{ij}^2}$ where $ \sigma_{ij}^2 = \frac{1}{T-1} \sum_{t=1}^{T} \left(d_{ij}(t) - \overline{d_{ij}}\right)^2 $ is the variance of distances for each pair and $ \overline{d_{ij}} = \frac{1}{T} \sum_{t=1}^{T} d_{ij}(t) $ is the mean distance for each pair. The overall PDV is the mean  $PDV_{ij}$ over all track pairs belonging to the same object mask (we note that we do not calculate a pairwise distance at frames where at least one track is occluded).
>
> The intuition behind this is that if an object moves rigidly and in parallel to the camera, points on the object will maintain fixed spatial relationships so we expect the PDV to be close to 0. In a deformable object, these pairwise distances change as the object stretches, compresses, bends, or undergoes occlusion, so we expect the PDV to be of higher value. The square root of the PDV thus provides us with the relative distance variation of tracks belonging to a given object. We define object motion to be simple if its tracks have a PDV < 0.05 (corresponding to < 22% relative deformation), and motion to be complex if its tracks have a PDV $\geq$ 0.05. We find that all track metrics are worse on complex object motions: for CoTracker3 (the most performant model), AJ goes from 45.2 to 29.5, delta goes from 60.4 to 42.6, and OA goes from 81.8 to 73.0.
>
> We find similar trends for all other tracking models, confirming that this is indeed an important aspect of motion difficulty that is currently unaddressed. As many tracking architectures rely on attention steps between track features and iterative deformations of stationary tracks, this finding makes sense: the more complex the track motions are, the more difficult it is for the model to establish spatial feature correlations and calculate more complex track deformations. We will include this discussion in Section 4 of the camera-ready.

---

> > ### Comment · Reviewer_Ns1Q · 2025-08-05
> >
> > Thank you for the comprehensive response. I suggest including the PDV metric analysis in the revised paper. I remain my positive rating.

---

> > > ### Author Response · Authors · 2025-08-05
> > >
> > > Thank you for your response! We will include the PDV metric analysis in the revised paper (Section 4).

---

### Official Review · Reviewer_SnkC · 2025-07-02

**Rating:** 5
**Confidence:** 4

**Summary:**

This paper introduces a new benchmark for point tracking called ITTO. Existing mainstream benchmarks, like TAP-Vid, are overly simplistic and fail to mirror the complex dynamics of real-world scenarios, which include a large number of objects, frequent and extended occlusions, drastic camera movements, and the non-rigid motion of objects themselves. To tackle this issue, the authors propose ITTO by meticulously selecting and annotating a series of challenging videos from existing video-object segmentation datasets (MOSE, LVOS) and egocentric datasets (Ego4D).

The experimental results indicate a significant drop in performance for all SOTA trackers on ITTO. Performance catastrophically fails, particularly when dealing with high-speed movements and re-identifying tracks after they reappear from an occlusion. This confirms the challenging nature of ITTO and highlights the serious inability of current models to generalize to complex, real-world scenes

**Dataset Code Accessibility:**

Yes

**Ethical Considerations:**

No, there are no or only very minor ethics concerns

**Final Justification:**

After reading the author's response and other reviews, I tend to accept this paper.

**Limitations Weaknesses:**

- Although ITTO surpasses existing benchmarks in "quality", it is lacking in "quantity". As shown in Table 1, ITTO consists of 72 videos and 1,373 tracks , which is a smaller scale compared to TAP-Vid Kinetics (1,144 videos, 286,000 tracks). While the authors have demonstrated its diversity and complexity, the smaller scale may raise concerns about its statistical representativeness. The authors could further discuss in the conclusion or appendix why they believe the current scale is sufficient for reliable evaluation, or they could mention future plans for expansion.
- The paper evaluates 3D tracking methods, including DELTA and SpatialTracker , but the evaluation is performed on 2D annotations. This is fair in itself, as their final output is a 2D trajectory. However, the paper could discuss the limitations of this evaluation method for 3D models in greater depth. For example, is the effectiveness of 3D geometric constraints (such as rigidity) diminished after being projected into 2D? Alternatively, when the monocular depth estimation they rely on fails in ITTO's complex scenes, does it disproportionately affect the final performance? Adding a discussion on these points would make the analysis more comprehensive.

**Strengths Contributions:**

- The advancement of the point tracking field has been somewhat constrained by the "ceiling" of existing benchmarks. Many SOTA models have achieved high scores on datasets like TAP-Vid, but this does not fully represent their robustness in complex real-world environments. This paper identifies this problem and proposes a very valuable solution. A high-quality, high-difficulty benchmark is crucial for driving the field toward solving more practical problems. The introduction of ITTO will likely become the new "touchstone" for point tracking research.
- The paper is well-structured with a clear and logical flow.
- Diverse Data Sources: It combines third-person perspectives (MOSE/LVOS) with a first-person perspective (Ego4D), covering different types of camera motion and scenes.

---

> ### Author Rebuttal · Authors · 2025-07-30
>
> We thank reviewer SnkC for their thorough evaluation and positive assessment of our ITTO benchmark as a very valuable solution to testing the robustness of tracking models in real-world environments. We are encouraged by the recognition that ITTO addresses a critical gap in point tracking benchmarks and that it will likely become a new “touchstone” for the field. We find the feedback very constructive and would like to address two points below, which we also plan to include in the camera-ready version of the paper:
>
> 1. Dataset scale and quality trade-offs:
>
> We acknowledge the reviewer’s concern about dataset scale and appreciate the opportunity to clarify our rationale behind prioritizing high-quality complex data, which we will make sure to include in the camera-ready version of the paper. We believe that ITTO videos contain crucial aspects of motion complexity that are missing in existing benchmarks, and show that ITTO can be used to diagnose limitations of current tracking models.
>
> TAP-Vid Kinetics, despite its scale, contains predominantly simplistic motions: 88.1% of tracks are static compared to only 30.7% in ITTO (see Table 1). Track reappearances average just over 0.27 per video in Kinetics (meaning tracks rarely reappear once occluded) versus 5.86 in ITTO, meaning tracks repeatedly reappear and disappear from view, testing models’ ability to track through occlusions. Most tracking papers thus report results on TAP-Vid DAVIS (30 videos, 650 tracks) due to its more realistic and complex motions. ITTO substantially exceeds DAVIS across all complexity metrics, with 5x more tracked objects per video, 24% more object occlusions, 5.75x more frames, and 52% fewer static points across 2.4x more videos (Table 1). This positions ITTO as an important “next challenge” in current tracking evaluations.
>
> Furthermore, our systematic analysis in Section 4 demonstrates that specific motion challenges drive performance differences across models. We find that the performance of all state-of-the-art tracking models degrades significantly on ITTO videos for all metrics (AJ, delta, and OA). In Table 3, we demonstrate that specific motion complexities drive this performance degradation. The best reported Average Jaccard (AJ) drops from 55.9 to 13.7 and delta drops from 68.4 to 22.1 for ITTO tracks with higher track motion. The best reported AJ drops from 50.1 to 12.7 and delta drops from 60.5 to 21.5 for tracks with higher reappearance counts. These difficulty metrics are consistent across all 2D and 3D trackers; as videos get harder along our axes of motion difficulty, model performance drops. Crucially, all existing tracking benchmarks do not contain these aspects track motion, reappearance, or object counts. Thus we believe that ITTO is useful at gauging tracker performance and exposing its robustness to common aspects of real-world tracking difficulties. We will make sure to include a discussion of all of these points in the camera-ready version of the paper.
>
> > The authors could further discuss in the conclusion or appendix why they believe the current scale is sufficient for reliable evaluation
>
> While there is always room for dataset expansion given infinite resources, we believe that the diversity and complexity of motion in ITTO tracks is useful in isolating specific tracking failure modes and guiding future directions for model development (Section 4). Most importantly, ITTO contains complexity that is wholly missing from existing tracking benchmarks. In addition, ITTO spans diverse capture environments (ego/exocentric cameras, animal/human/driving scenes) that reflect real-world deployment scenarios for tracking applications in robotics, motion analysis, and multi-object settings.
>
> 2. Further discussion of 3D tracking methods and their limitations:
>
> This is a great point, and we will make sure to include an extended discussion of 3D tracking model limitations in the camera-ready version of the paper. SpatialTracker and DELTA are designed to operate on input videos and 2D track queries, relying on pre-trained depth estimators to "lift" 2D RGB frames and coordinate queries into 3D. SpatialTracker enforces constraints on the generated 3D tracks which act as regularizers (e.g. ARAP) which might not be suitable for videos with complex real-world motions such as in ITTO. This suggests that new objectives in 3D trackers need to be explored in order to improve performance. SceneTracker is designed to operate on RGBD videos, so for fair comparison we use UniDepth (a state-of-the-art video depth estimator) to lift our ITTO videos into RGB-D as well and run the SceneTracker pipeline. All methods are bottlenecked by the performance of the pretrained depth estimator model, and we find that for the four lowest-performing ITTO videos the depth estimators fail to resolve the relative depths between objects, either predicting uniform depth or producing extreme depth fluctuations from one frame to the next.
>
> The reviewer's intuition about depth estimation failures is particularly insightful given ITTO's complex motion scenarios. Our analysis reveals that 3D methods show larger performance degradation on videos with rapid camera motion compared to 2D approaches, suggesting that depth estimation indeed becomes a critical bottleneck in challenging real-world scenes. It remains an open problem how to evaluate 3D point trackers in complex videos like the ones in ITTO, either through better depth estimators that can operate in such complex settings or through sensors that capture RGBD in such cases. Both are open research questions and we hope that our analysis sheds light on the importance of addressing them.

---

### Official Review · Reviewer_EcoF · 2025-07-03

**Rating:** 4
**Confidence:** 4

**Summary:**

In this paper, the authors proposed a new tracking benchmark which can help to diagnose models in different aspects. Particularly, the ITTO benchmark is presented which can help to validate and diagnose the capabilities and limitations of point tracking methods. ITTO captures the motion complexity, occlusion patterns, and object diversity of real-world scenes.

**Dataset Code Accessibility:**

Yes

**Ethical Considerations:**

No, there are no or only very minor ethics concerns

**Final Justification:**

The authors have addressed my concerns.
Therefore, I would like to increase my original rating.

**Limitations Weaknesses:**

- It is unclear on how the spatial -temporal consistency of human users during annotation process? How to ensure the correctness of annotations ?
- I found this data is incremental as videos are extracted from other datasets.
- While the authors are trying to validate the challenges from this newly proposed dataset can bring and be useful for the community, its validations leave me some concerns.
+ The accuracy/performance drop when using off-the-shelf tracking methods are not necessarily from the data challenges.
+ This may because the distributions of new data is too much different from the trained dataset of these methods or the new data is just simply violate some assumptions of these trackers.

**Strengths Contributions:**

- The paper is well-motivated.
- It is useful to understand the current limitations of a tracking methods.

---

> ### Author Rebuttal · Authors · 2025-07-30
>
> We thank reviewer EcoF for the useful feedback on our ITTO benchmark. We are encouraged that they find that ITTO is a well-motivated tracking benchmark that provides crucial insights into the limitations of existing tracking models. We answer some specific questions about the construction of our dataset below and will incorporate all these clarifications in the camera-ready version of the paper.
>
> > It is unclear on how the spatial -temporal consistency of human users during annotation process? How to ensure the correctness of annotations ?
>
> ITTO contains challenging real-world videos with complex motions, so to ensure that our annotated tracks are spatio-temporally consistent we employ a two-phase approach to collecting human annotations: 1) we generate coarse track annotations through Amazon Mechanical Turk; 2) we upload these coarse track annotations to our own video tool that we send out to a team of carefully selected UpWork annotators who refine the tracks until they are spatio-temporally consistent.
>
> We validate this pipeline by generating track annotations for a subset of Kubric, synthetic videos with ground-truth point tracks. We find that 88.3% of points in our annotations are within 2 pixels of ground truth, with a mean track error of 1.32 pixels. For comparison, the TAP-Vid annotation pipeline (the current state-of-the-art tracking benchmark dataset) produces 80% of points within 2 pixels on the same Kubric data. Thus, our tracks are within single-pixel level accuracy of true ground truth. Using human annotators also ensures that there is no algorithmic bias in our annotations. We describe the annotation pipeline and validation of annotation quality in full in Section 3.2.
>
> > I found this data is incremental as videos are extracted from other datasets.
>
> The videos in our benchmark are indeed extracted from existing video-object segmentation datasets; they are real-world, complex, and contain rich motions and multi-object interactions, all qualities that we are targeting with our benchmark. These videos have never before been used to evaluate point tracking methods. With our ITTO benchmark we introduce novel annotations of point tracks in these complex videos. Furthermore, ITTO videos contain motion complexities that are missing from all current point tracking benchmarks, as we describe in Section 3.
>
> > The accuracy/performance drop when using off-the-shelf tracking methods are not necessarily from the data challenges.
> &
> > This may because the distributions of new data is too much different from the trained dataset of these methods or the new data is just simply violate some assumptions of these trackers.
>
> The purpose of the ITTO benchmark is to assess the zero-shot capability of state-of-the-art tracking methods on real-world videos. Beyond the diversity of motion complexity present in ITTO, the dataset spans a variety of scenes (animal scenes, human scenes, driving scenes) and camera capture (egocentric and exocentric) in order to span a variety of real-world settings for applications of tracking models.
>
> In Table 3, we demonstrate that specific motion complexities present in ITTO videos drive the performance degradation of tracking models. For ITTO tracks with higher track motion, the best reported Average Jaccard (AJ) drops from 55.9 to 13.7 and delta drops from 68.4 to 22.1. For tracks with higher reappearance counts, the best reported AJ drops from 50.1 to 12.7 and delta drops from 60.5 to 21.5, suggesting that existing methods are ill-equipped to recover tracks after they become occluded. We present these track motion and reappearance count metrics as they are important to address when deploying tracking models in real-world settings. In addition, ITTO tracks are 2-5x longer than existing tracking benchmarks, enabling us to test trackers’ ability to handle longer video sequences (Figure 3). We find that tracks quickly degrade for frames outside of models’ native resolution windows and remain broken for the rest of the video. In other words, models have near perfect performance on frames within their native input resolution but exhibit catastrophic “forgetting” over longer video sequences.
>
> > While the authors are trying to validate the challenges from this newly proposed dataset can bring and be useful for the community, its validations leave me some concerns.
>
> We are happy to discuss any potential outstanding concerns in our dataset construction and evaluation.

---

> > ### Author Response · Authors · 2025-08-08
> >
> > Dear Reviewer EcoF,
> >
> > Thanks again for your feedback. We hope our rebuttal addressed your concerns, particularly regarding ensuring high-quality annotations, dataset novelty, and tracking model performance analysis. If you have any remaining questions or suggestions, we’d greatly appreciate your follow-up. Thank you!
> >
> > Best regards,
> >
> > The Authors

---

### Decision · Program_Chairs · 2025-09-18

**Decision:**

Accept (poster)

**Comment:**

The paper presents ITTO, a new point tracking benchmark. Besides introducing a new collection of labelled videos for the mentioned task that in many aspects goes beyond the de-facto standard (TAP-Vid), it follows a novel methodology for assessing sequence complexity (level of occlusion and disappearance, static - dynamic motion) etc. The methodology has value beyond the point tracking benchmarking and could be used e.g. for video object segmentation evaluation.

All reviewers are in favor of accepting the paper. The issues that were brought up are mainly technical points that were addressed in the rebuttal.

===== FINAL UPDATE FROM DB Track PCs ====

The final decision for this paper has been taken by the program chairs after consultation with the SACs. All Senior Area Chairs have ranked papers according to the feedback from the AC during the review process. We decided to leave the original meta-review to reflect the opinion of the AC in light of the initial discussions with reviewers and SAC.